# Portraits of Working Women: Lola Ridge's "The Ghetto" and the Visual Record

Linda Arbaugh Kinnahan

Faculty of English, McAnulty College and Graduate School of Liberal Arts, Duquesne University, Pittsburgh, PA 15282, USA; kinnahan@duq.edu

**Abstract:** This essay focuses on Lola Ridge's long poem "The Ghetto" in relation to the gendered imagery and visual construction of the modern laborer emerging across early twentieth-century print media. Perpetuating gendered notions of the modern worker as predominately masculine, late nineteenth- and early twentieth-century visual representations of the laborer typically feature manly, virile figures, often in resistance to capitalism and inevitably eliding the industrial woman laborer. Ridge's "The Ghetto" alternatively locates modern labor in the female industrial worker. The essay considers the poem's splicing of collective and individual portraits of immigrant working women, developing a visual rhetoric that asserts women's agency amidst modernity's changing forms of work, insisting upon their visibility as workers, activists, and feminists. Consideration of several visual print genres includes women's labor publications; social and industrial documentary photography; and periodical illustrations from *The Masses*. In visually representing women workers, these sources of visual media contextualize Ridge's approach in "The Ghetto" and social attitudes toward gender and labor persisting in the century's early years.

**Keywords:** women's labor movement; ghetto; social documentary photography; Lola Ridge; John Sloan; Frances Benjamin Johnston; Lewis Hine





## 1. Introduction

A spectacle of modernity churning with commercial and industrial energies fueled by swelling populations of immigrants, enduring through generations of mothers, and agitated by young laborers keen on change, the urban scenes in Lola Ridge's modernist long poem "The Ghetto" vividly depict New York's Lower East End in the first two decades of the new century. Upon this screen of the modern, the poem projects the image of the industrial woman worker. As "All day the power machines/Drone in her ears" at the garment factory, the figure of female labor distills the poem's argument with capitalism in precise and visual terms, pitting her against industrial systems organizing laboring bodies for the purposes of profit (Ridge 1918, p. 6). While the poem weaves interrelated strands of early twentieth-century life in the Jewish Lower East Side, "The Ghetto" is distinct from contemporaneous depictions—visual, literary, and journalistic—of this rapidly changing urban scene in focusing poetic attention on industry's female laborers as central to modernity and as class-specific exemplars of the "new woman", a category typically overlooking the working class.[1] Indeed, the public imagination of the "laborer" often elided women, or rendered them passive and victimized, despite the visible activism of the women's labor movement in the century's first fifteen years. In large part, the image of the laborer remained male for both supporters and critics of workers' movements, perpetuating gendered ideas associated with wage labor from the nineteenth century's shift to corporate capitalism in America and its changing forms of labor.

Women's growing entry into the workforce after the Civil War moved more of them out of the household and beyond areas of paid work typically limited to agricultural or domestic spheres. As Maurine Weiner Greenwald describes,

The birth of the modern corporation marked the beginning of important changes in the location and nature of women's paid labor. The reorganization of capital into corporations facilitated the development of large-scale manufacturing, the national distribution of goods, and the growth of a host of associated commercial enterprises . . . . [N]ew and expanded enterprises created thousands of jobs for women. (Greenwald 1984, p. xv)

Typically, women were regarded as cheap labor, and strict gender segregation denied them jobs open to men and equal pay.

The legions of women entering the industrial labor market, especially by and after 1890, derived large numbers from immigrant groups arriving from southern and eastern Europe. In discussing "The Ghetto", Cristanne Miller reminds us that from 1880 to 1920, approximately 28 million immigrants entered the U.S., "the vast majority settling initially in New York City", with the "largest group from the Russian empire (mostly Jewish) and the second largest group from Italy (mostly Catholic), both groups typically settling first in the cheap tenements of the Lower East Side"; the Jewish population comprised 31 percent of the city in 1910 (Miller 2007, p. 458). Viewed with suspicion by native-born Americans as well as Protestant Anglo and northern European immigrants, these new arrivals brought different religions and customs feared as unassimilable and even unhealthy. Hierarchies of job opportunities, most often class- and ethnically-based, limited job options for these immigrant women and their daughters, pressed by the economic need to work but resigned to wage-jobs such as factory workers or domestic workers rather than the more prestigious professions of nursing, teaching, or even department store work, which all retained qualities deemed feminine and less threatening to gender norms. As Alice Kessler-Harris documents, "By the turn of the century, from one-third to one-half of the populations of major cities . . . consisted of poor immigrant families", in which the income of a "single wage-earner" was "frequently impossible" for a family to live on (Kessler-Harris 1982, p. 121). Mothers, daughters, wives, and single women entered the labor market outside the home, as a "steadily rising number of women, married and unmarried . . . felt impelled to contribute to their families' economic sustenance. Among teenage girls, three-quarters or more probably sought paid work by 1890", and by 1920, the "percent of married women working for wages" nearly tripled since 1890 (Kessler-Harris 1982, p. 122).

Ridge's focus on women from immigrant Jewish groups interweaves dimensions of class, gender, and ethnicity within visually rich urban and industrial scenes. In presenting the American worker as female, "The Ghetto" stands alone as an American early modernist long poem and speaks more broadly to the ways in which the female worker, particularly the industrial woman laborer, is imagined and constructed in an increasingly visual print environment. Ridge's image-based poetics in "The Ghetto" and—comporting with her earlier training as an artist—her poem's attentive visual apprehension participate in modern culture's proliferation of the visual as a mode for shaping ideas and making meaning. In a time of intensifying labor agitation between 1890 and 1920 and changing conventions of gender, how might we understand Ridge's woman-centered portraits interacting with the visualization of modern labor and the impact of the women's labor market in a media environment increasingly utilizing visual images? How might we consider the visual rhetoric developing around women's wage labor? Thinking through the relatively little visual attention paid to women's labor in relation to the figure of the male laborer, in conjunction with socio-cultural anxieties about women's work outside of the home and the spectacle of women's strikes and political protests, this essay turns to distinctive print environments that did participate in generating a visual lexicon of the modern laborer as female.

Lola Ridge has suffered critical neglect for decades, and her poetry's political proclivities toward socialism, anarchism, and feminism surely helped bury it. Recent critics, notably Nancy Berke, Tim Newcomb, Cristanne Miller, Michelle Gaffey, Caroline Maun, and Danielle Tobin have brought renewed attention to her work (and brought portions of it back into print), and a comprehensively rich biography by Terese Svoboda joins a recent

annotated edition of the 1918 *The Ghetto and Other Poems*, edited by Lawrence Kramer. As Ridge's first book, *The Ghetto and Other Poems* presents the Lower East Side of New York City through its markets, commerce, and labor while extending a critique of capitalism to consider the plight of workers in America and other issues of social injustice.[2] The book's several titled sections begin with the long poem "The Ghetto", followed by other poems depicting Manhattan and its inhabitants, poems advocating workers' rights and protesting social injustice, and several lyric meditations.

Active within poetic, intellectual, and activist circles in New York in the first decades of the twentieth century, Ridge's leftist and feminist convictions shaped her modernist experiments with the long poem and lyric forms and influenced her editorial work with the important avant-garde journals *Others* and *Broom*. Hosting salon gatherings in the late teens and early twenties attended by poets including Marianne Moore, Mina Loy, and William Carlos Williams, as well as social justice advocates such as Dorothy Day (committed to justice issues before conversion to Catholicism), John Reed, and Floyd Dell, Ridge first distinguished herself in the 1910s as a modernist "proletarian poet", attentive to poverty, injustice, and capitalist abuses.[3] Publishing in *The Dial* and *Poetry* (including poems of socialist protest), Ridge also placed poems in "left-leaning" magazines and publications such as *The New Masses*, *The New Republic*, and *The New York Call*, and Emma Goldman included two of her poems in *Mother Earth*, later calling her "our gifted rebel poet" (in Svoboda 2016, p. 71). *The New Republic* published sections of her long poem "The Ghetto", "the first English-language modernist long poem on the subject of ghetto life to be published in the United States" (Berke 2010, p. 34).

Ridge arrived in New York in 1908 amidst myriad and related upheavals that unsettled American social hierarchies, including dramatic waves of immigration, full-blown labor activism in streets and meeting halls, Progressive Era socio-economic reforms, and an invigorated feminist movement focused diversely on political, social, sexual, and labor rights for women. Having immigrated from Ireland to New Zealand (as a child) to Australia and then to San Francisco before landing in New York, Ridge settled among Lower Manhattan's more recent immigrants from eastern and southern Europe, focusing much of "The Ghetto" on Jewish communities, families, and workers. Ridge's own immigrant status grounds her subsequent involvement with the city's immigrant population and her poetic rendering of their experiences, especially those of women. She was drawn to the Ferrer Center, which opened in 1910 to serve as a "community center for anarchists and freedom-loving writers and artists" that attracted and welcomed immigrants interested in anarchist and socialist ideas (Svoboda 2016, p. 75). In addition to managing the center for a period, Ridge helped run its Modern School, founded to educate children through progressive philosophies and host to poetry readings and political gatherings, editing its publication *The Modern School* in 1912 (Miller 2007, p. 466). She developed relationships with fellow revolutionaries such as Emma Goldman and John Reed, along with a range of artists, writers, and activists.

Leaving the city from 1912 to 1918 to live nomadically around the country with anarchist David Lawson, her future husband, Ridge continued to write the poems that would be included in her first book. She "often wrote about labor" and political events, responding to the activist Tom Mooney's arrest, the race riots in East St. Louis, and the murder of labor leader Frank Little (Svoboda 2016, p. 97). Returning to New York in 1918, she published several poems right away in literary and progressive venues (including *The Dial*, *Others*, and *Poetry*), placing the long poem "The Ghetto" in *The New Republic*. The poem's focus on immigrant life in Lower Manhattan attracted the publisher B. W. Huesch (himself a Jewish immigrant who had also published Joyce and Lawrence), who offered her a book contract. Appearing in September 1918, the book "created an immediate sensation", especially for its depiction of immigrant life (Svoboda 2016, p. 98).

"The Ghetto" celebrates the diaspora of Jewish "old world" arrivals and their progeny, capturing a cross-generational sense of the clashes, continuities, and strivings energizing this community. Women and their labor—domestic, commercial, and industrial—center the poem's vision of the ghetto. Each of the nine sections details some aspect of life in

the "Ghetto"—identified as the Lower East Side of New York City—moving in and out of street scenes (markets, a parade, streaming foot traffic), homes, workplaces, and cafes. The speaker boards in a tenement flat and the poem oscillates between her observational point of view and other registers of perception, engaging camera-like techniques of close-up, panorama, or bird's-eye perspectives, and more impressionistic, associative shifts.

Panning the cross-generational scene of Hester Street, the poem opens with the summer heat pressing upon people in the streets and markets. Specifying the mothers and their babies as well as the younger women in groups who move in and among the crowds, this contemporary scene counterpoints imagined memories of Jewish "ancient" mothers, whose vision and strength sustain future generations, an image of the diaspora rooted in maternal ancestry. These images of the many, evoking both past and present, lead to a set of individualized portraits in the second section, as the speaker emerges to describe three of her fellow female boarders, all garment workers, who represent a younger generation of immigrants, often sharply at odds with the older customs of their parents. Sadie, Sarah, and Anna also embody the female laborer.

Remaining sections build upon the collective and individualized portraits of the ghetto that initiate the poem. Subsequently, in the third section, the speaker encounters and tries to befriend an immigrant child at a patriotic parade before seeing her run away in fear of the noises that remind the girl of the pogroms her family had fled. Wandering the market stalls past different vendors, the presumed speaker (the "I" has receded for a more camera-eye recording of detail) in the fourth section lists a cascade of goods for sale amidst a "crazy quilt" of contrasts between old and young, between mothers with children, and ethnic merchants with customers dealing in the shadow of distant skyscrapers (Ridge 1918, p. 12). The speaker re-emerges in the fifth section, watching from her window the morning routines and evening rituals of Jewish neighbors, connected by Sabbath lights in windows. A neighborhood café, in the sixth section, hosts older customers intruded upon by young working-class men with their "committee" leaflets, who meet (with some women present) to argue socialist and anarchist politics in the seventh section. By evening, in the penultimate section, as the streets and markets shut down, the ghetto becomes a darkened cramped "ova", suggesting its regenerative energy amidst the commotion and crowdedness (Ridge 1918, p. 22). This energy bursts forth in the final section, as the speaker enters her room at dawn and life begins again. The poem ends with a cascade of enlivened stanzas hailing the life forces of the ghetto, the "Astounding, indestructible/Life of the Ghetto" (Ridge 1918, p. 26).

Focusing its discussion of women's labor through a primary consideration of the first two sections of the poem, this essay takes its cue from the alternating portrayals of woman-centered crowds and female individuals with which Ridge frames the rest of the poem. The following discussion focuses upon women's collectivity in "The Ghetto" and in the slightly earlier poem "Bread", discerning how the imagery and language of each poem respond to and participate in representing the modern spectacle of women's mass presence in parades, pageants, protests, and strikes during the first years of the twentieth century. Such public events increasingly sought and brought visibility to the activists and groups pressing labor and equal rights demands (sometimes separately, sometimes in concert). Garment workers led the wave of strikes and activism by women laborers. Reading Ridge's portraits of individual garment workers, the final two sections of this essay consider the visible construction of the industrial woman worker and her labor: to what degree was she visually represented, and in what contexts? What aspects of print culture circulated images of modern labor performed by women, and how so? How do gender associations shape visual vocabularies developing around the modern woman worker? How are social anxieties about women's work outside the home conjoined with stereotyping of new immigrants?

Approaching these questions, the essay closely reads the first and second sections of "The Ghetto" while choosing to thicken the contextual materials that Ridge's poem variously evokes, responds to, or corresponds with in subject matter and parallel strategies.

The essay's final two parts discuss specific genres of print culture that visually represented the industrial labor of women and the laborer as female, albeit through a lens shaped by differing concerns and objectives. Tracing a modern diversification of visual cultures around the figure of the working woman, this essay turns from a brief survey of post-bellum print media to distinct genres of the new century: women's labor publications; social and industrial documentary photography (in the cases of Jacob Riis, Lewis Hine, and Frances Benjamin Johnston); and periodical illustrations for the radical press, in the sympathetic examples of artist John Sloan.

## 2. Visualizing Collectivity

Amidst an increasingly image-based culture, Ridge's modernist depictions of the city shape a visual aesthetic that critics have insightfully linked to modern conditions and technologies. Nancy Berke compares the poem's opening stanzas to cinematic film and, evoking photographic technologies new to a popular audience, she describes the poem's structure as sequences of "imagist snap-shots that show fragments of immigrant Jewish urban experience" and develop multi-dimensional "portraits of working people" (Berke 1999, pp. 71–72).[4] These "snapshots of lower Manhattan's immigrant enclaves" and the "highly visual prose feel of her verse" produces an "urban sketch" from the perspective of a "poet/flaneuese of modernist New York." Regendering the figure of the flaneur, Ridge embodies the "female artist who engages and critiques the modern city from her position as a new American" and imagines "a female presence within decidedly masculine spaces" (Berke 2010, p. 31). Later poems in the book, such as in the "Manhattan Lights" section, have a "fragmented snapshot quality", Berke notes, reflecting the rapid movement and change of the modern metropolis (Berke 2010, p. 39). Certainly, "The Ghetto" begins in cinematic fashion, capturing close-up shots of Hester Street inhabitants to then zoom out upon a panoramic view of the urban market thoroughfare packed with close-pressed bodies, stressing images of the city's "tide of flesh" and "surges of flesh." This flesh issues from women's bodies as an "abiding/Brood", linking their bodies to an endurance and quest for freedom carried from the past (Ridge 1918, p. 4).

Women infuse the picture of the Lower East Side introduced in "The Ghetto" and remain particularly central in the second section, which introduces the speaker as a woman boarding and working among Jewish immigrant women in the garment industry. The poem's pronounced attention to these women, and to the broader context of immigrant women employed in manufacturing work, reshapes dominant visual rhetoric depicting the modern laborer as masculine and counters the elision of the female industrial worker from such depictions. Perpetuating traditional gendered notions, late nineteenth- and early twentieth-century visual representations of the laborer typically featured manly, virile figures who, in radical and labor presses, embodied a resistance to capitalism in their manliness. Ridge's "The Ghetto" alternatively locates resistance in the female industrial worker, splicing together individual and collective portraits of immigrant working women, developing a visual rhetoric that asserts women's economic agency amidst modernity's changing forms of work, and insisting upon their visibility as workers, activists, and feminists.

Financially strapped in the early years of the century, Ridge lived near Hester Street in Greenwich Village, where she interacted with an immigrant population diverse in language, religion, and ethnicity, whose daughters populated the garment industry's workforce.[5] Alongside immigrant women, Ridge—herself an Irish-born and New Zealand-raised immigrant—worked in factories on the Lower East Side for a period. Attentive to the rise in women's labor activism, Ridge's socialist and feminist sensibilities shape her poetry of the teens and twenties, and her early years in New York City coincided with the intensified public presence of masses of women laborers striking for better conditions and large gatherings of women's rights protestors, a collective presence that engaged public visibility and spectacle in the century's first decades. Through language evoking visible masses (a "tide" of bodies that "surge" on the streets) and denoting women's dominant

presence, the opening section of "The Ghetto" introduces a panoramic view of the area around Hester Street in the Lower East Side, where market vendors and shoppers crowded the streets in the early years of the twentieth century (Figure 1). Imagined as "a feminized space", the ghetto comes into view as "urban spaces upon which women encroach: the street, the market, the sweatshop", juxtaposing "domestic spaces" and "spaces reserved for commerce" that call "attention to Jewish women's commercial roles" (Berke 2010, "Electric Currents", p. 37).

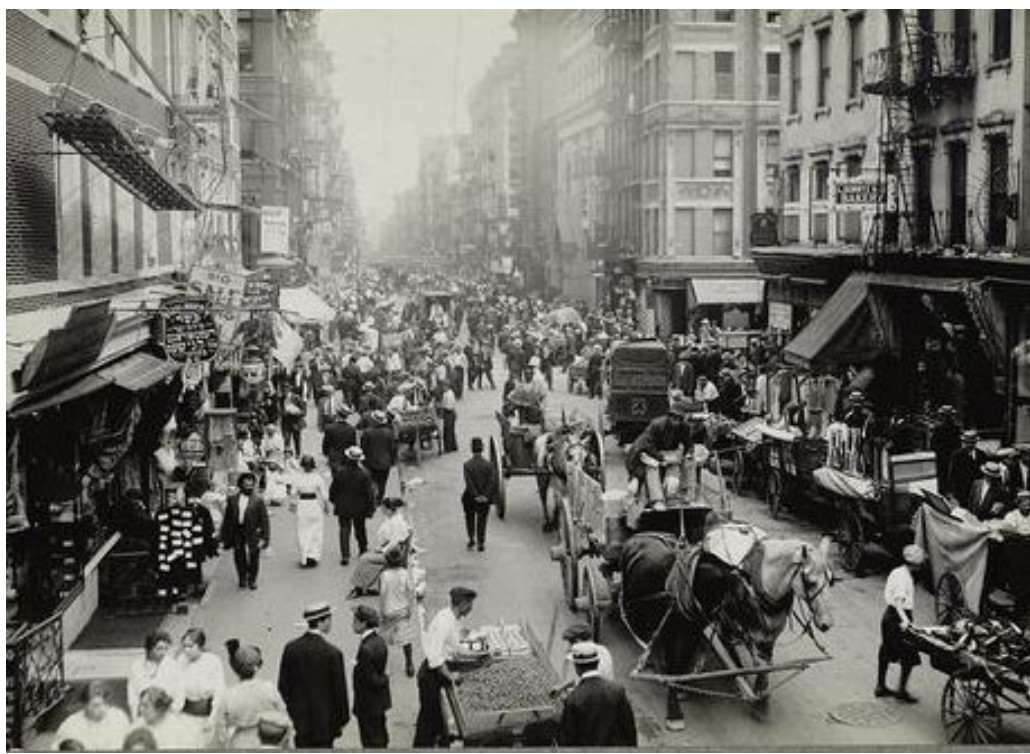

**Figure 1.** Lewis Hine, 1912. Market Day in Jewish Quarter of East Side, New York City. New York Public Library Digital Collections. Available online: https://digitalcollections.nypl.org/items/510d4 7d9-4da8-a3d9-e040-e00a18064a99 (accessed on 29 August 2022).

This imagistic depiction of economic agency for women, inseparable in the poem from women's historical roles as caregivers and preservers of culture, evolves as groundwork for the political and social energies of a younger generation of working women. Introducing groups moving through the streets and meeting halls made up of women working in factories, "The Ghetto" captures a visual milieu made suddenly familiar in the new century by women's strikes, suffrage marches, and other woman-led mass protests (Figure 2). Newspaper and magazine reports on women's protests and strikes helped circulate images of women marching, speaking, and gathering at events such as the Uprising of the 20,000, led by the International Ladies' Garment Workers' Union with support from the National Women's Trade Union League (WTUL). Showing the force of women's union organizing, this shirtwaist factory strike of 1909 gathered the nation's largest number of female strikers up to that date, most of whom were Jewish women (Figure 3).

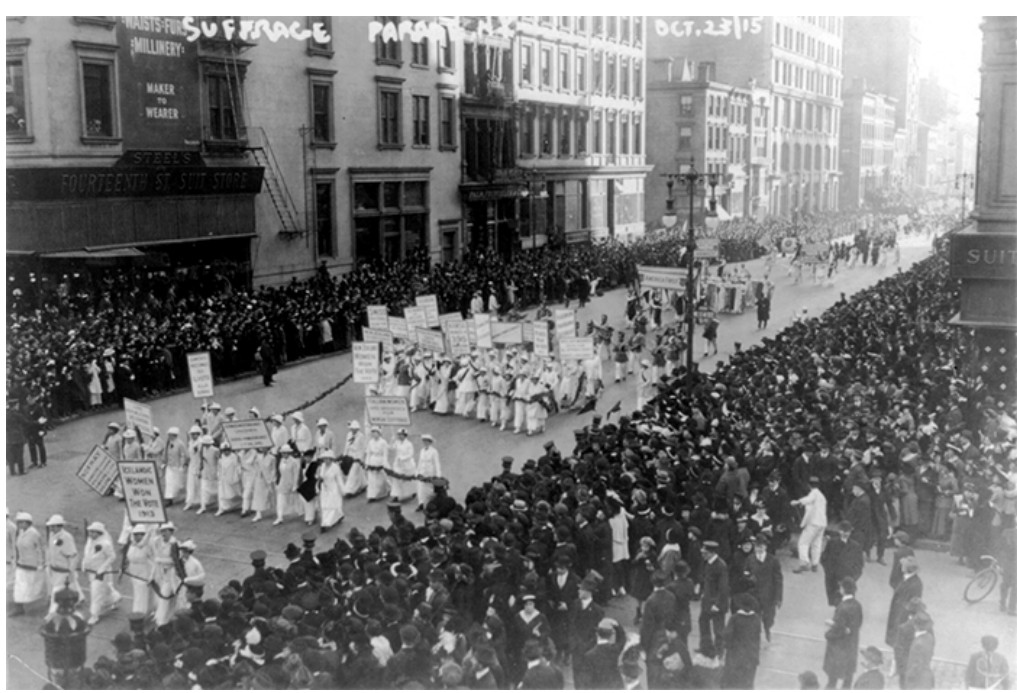

**Figure 2.** Women march for suffrage. 23 October 1915, New York City. Library of Congress.

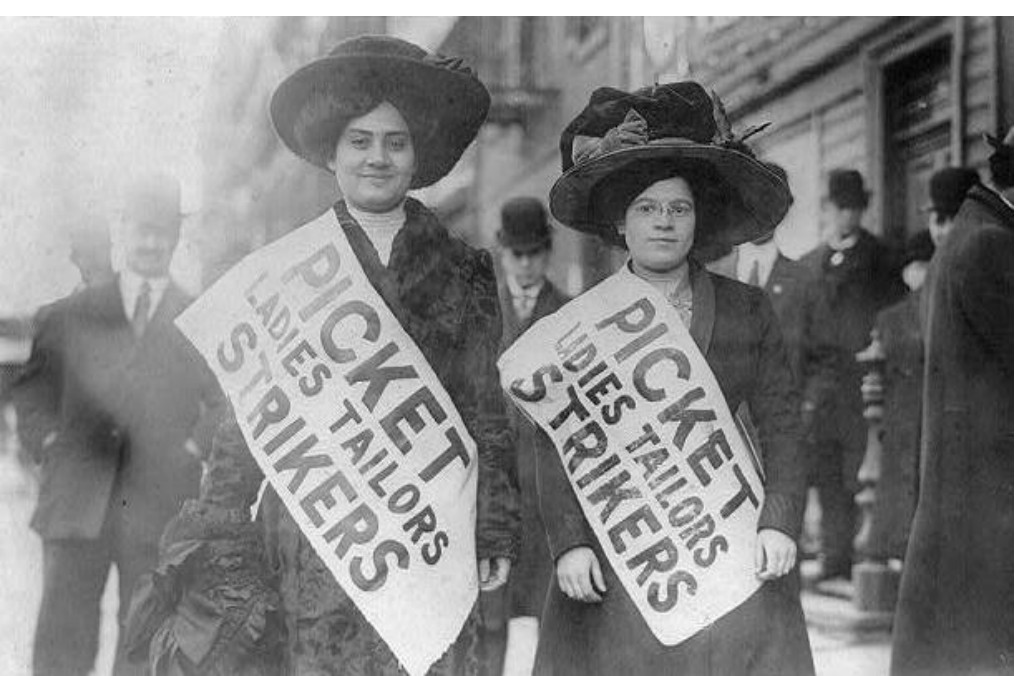

**Figure 3.** Women tailors on strike, Uprising of 20,000, 1909. Wikipedia Commons.

A wave of strikes across the country from the early 1900s through the Great War generated publicity for the causes espoused by women workers, largely excluded from male labor unions. Women gained visibility as organizers and participants in nationwide strikes orchestrated by new women's labor unions. "The Ghetto" pays particular attention to female garment workers, the largest group of unionized women and prominent in labor activism, and to Jewish immigrants. Along with Italian women, Jewish women representing new waves of immigrants from eastern and southern Europe assumed strong leadership in the women's labor movement amidst suspicion of the "foreign" influence of immigrants' Catholicism, Jewishness, and cultural differences, regarded by many as unassimilable to a

largely Anglo-Saxon and Protestant culture. Labor collectivity across ethnic lines became necessary—although not without conflict—as new women's unions formed, and as they navigated relations with middle-class women's rights groups for common causes, such as suffrage.[6] Coast-to-coast media attention to the combined efforts of women's labor and enfranchisement groups intensified around such events as the 3 March 1913 Women's Parade in Washington D.C. Held on the eve of Woodrow Wilson's inauguration amidst official promises of an unobstructed marching route made secure by police, the marchers were attacked by members of the crowd, with little police effort to protect the marchers. Lillian Carr, covering the march as a representative for *Life and Labor*, the magazine of the WTUL, stressed the involvement of women's trade unions and workers in this suffrage event while disparaging the city's failure to maintain order and stating that "evidently men do not like women to parade" (Carr 1913, p. 112). Photographs accompanying the article show the massive crowd (Figure 4).

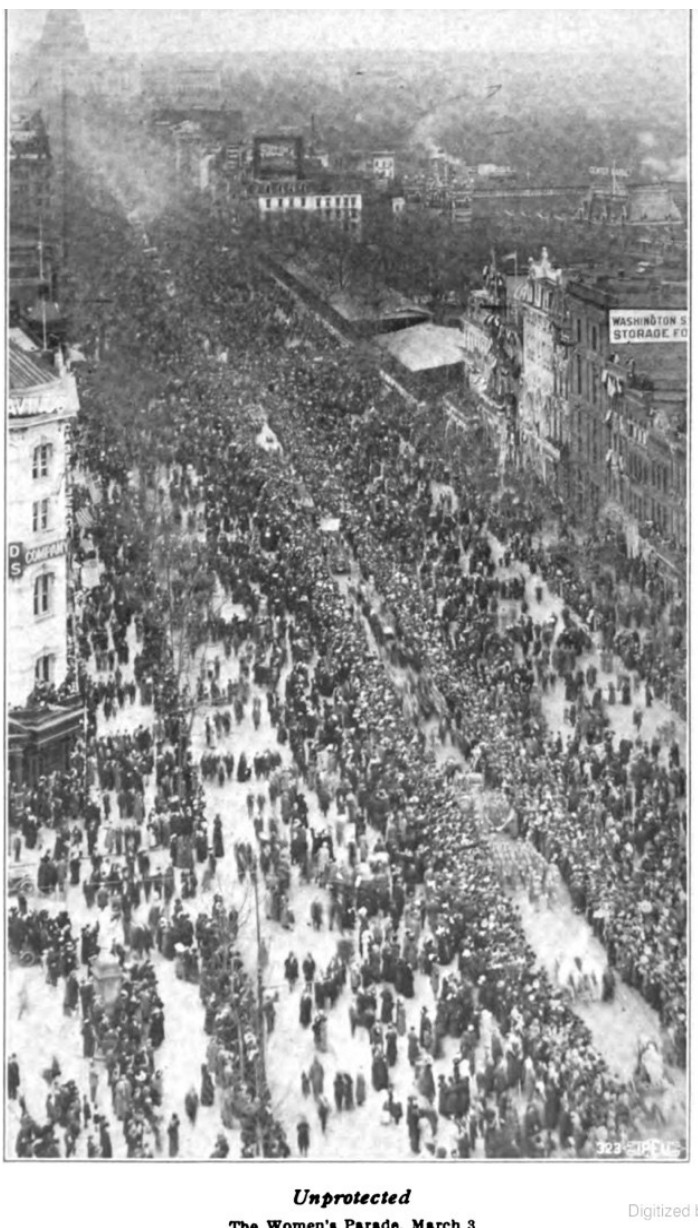

*Unprotected*
The Women's Parade, March 3

**Figure 4.** "Unprotected: The Women's Parade, March 3" in *Life and Labor*, April 1913, p. 114. Photo credited to *New York World*.

Adopting the visual phenomenon of women joined together in collective protest, Ridge's poem "Bread" appeared in the socialist *New York Call* on 19 October 1917, a year prior to *The New Republic's* printing of "The Ghetto." This anti-war poem recalls a demonstration by women at New York City Hall the previous December, protesting "against the high cost of living" and the bodily degradation of prioritizing financial gain over human need.[7] Critiquing the economy's harm to women and linking it to war, the poem insists upon women's active and *collective* response, differing from common depictions in the popular press of women driven to prostitution or starvation by the wage-labor system in representations of a solitary victimization without recourse. Alternatively, the poem voices the powerful impact of collective action as the women join and display their bodies to document evidence of injustice while serving as instruments of protest. Demanding "Bread! Bread!", the women sing

> [ . . . ] to a rhythm of upturned faces,
> White as under a flame,
> And a flashing of slender wrists
> And a swaying of unset hips . . .
> That desperate, hag-worn cry –
> Strange on the dewy lips. (in Van Wienan 2002, p. 193)

The women's "strange" and unnatural transformation from young women with "dewy lips" into poverty-marked "hag-worn" and "[s]hawled women" forms the crux of their protest, their rags transformed to the "bloodied gown" of the collective "flags" demanding that human needs be met. Their bodies directly connect the protest to human need, not only through their own deprivation but also explicitly through the image of their connection to nourishment, "their thumbs upon the rye" and "their thumbs upon the wheat" (in Van Wienan 2002, p. 193).

Combining imagery of bread and motherhood with calls for collectivity recalls labor activists' own language, especially in demanding better wages for women. Women labor leaders refuted the then-standard notion that the working man is the family supporter, as had Leonara O'Reilly for a decade when she claimed in an article for *The Union Printer* in 1917 that the "antique notion that man is the sole producer of wealth" defies the reality of "those 800,000 women in our own state who tread the lock step through life for their daily bread" (in Vapnek 2009, p. 160). Evoking a massive gathering of women workers, O'Reilly advocated better wages within a restructured paradigm of labor she termed "social motherhood"—a "care for others rather than the profit principle", and a "social recognition of women's work of caregivers as well as wage earners" (Vapnek 2009, p. 160).

While Ridge's imagery, like O'Reilly's words, presents women as agents of change, willing to display their bodies as evidence and protest but unwilling to accept a passive state of victimization, the 1917 readers of "Bread" would be familiar with a plethora of alternative media approaches that more often staged women as victims of market forces. Consider an example from the radical press depicting impoverished women as victims needing to be saved. *The Masses*, well known to Ridge, devoted its December 1911 issue to a "Woman's Number." While the issue's cover centers a three-quarter facial portrait of an attractive young woman, the frontispiece illustration, by A.O. Fisher, is direr (Figure 5). Depicting a group of women in a line extending beyond the frame of the page, the image suggests multitudes from across generations standing outside an impersonal brick building—perhaps a factory—seeking work as they huddle grim-faced under shawls. The caption underneath the women reads "The Cheapest Commodity on the Market", introducing a one-page editorial on the facing page, bearing the same title as the caption, that emphatically argues "To be a woman in modern capitalist society means to be the cheapest commodity on the market", reflecting the reality of women's lower wages (The Cheapest Commodity 1911, pp. 4–5). Rallying for women by presenting them, through text and visual illustration, as victims of capitalism rather than as agents of change, the editorial justifies its outrage through conventional associations of womanhood and motherhood, for

"From these women will come the race of the future", and "the degradation of women [is] the degradation of the whole race" (The Cheapest Commodity 1911, p. 5).

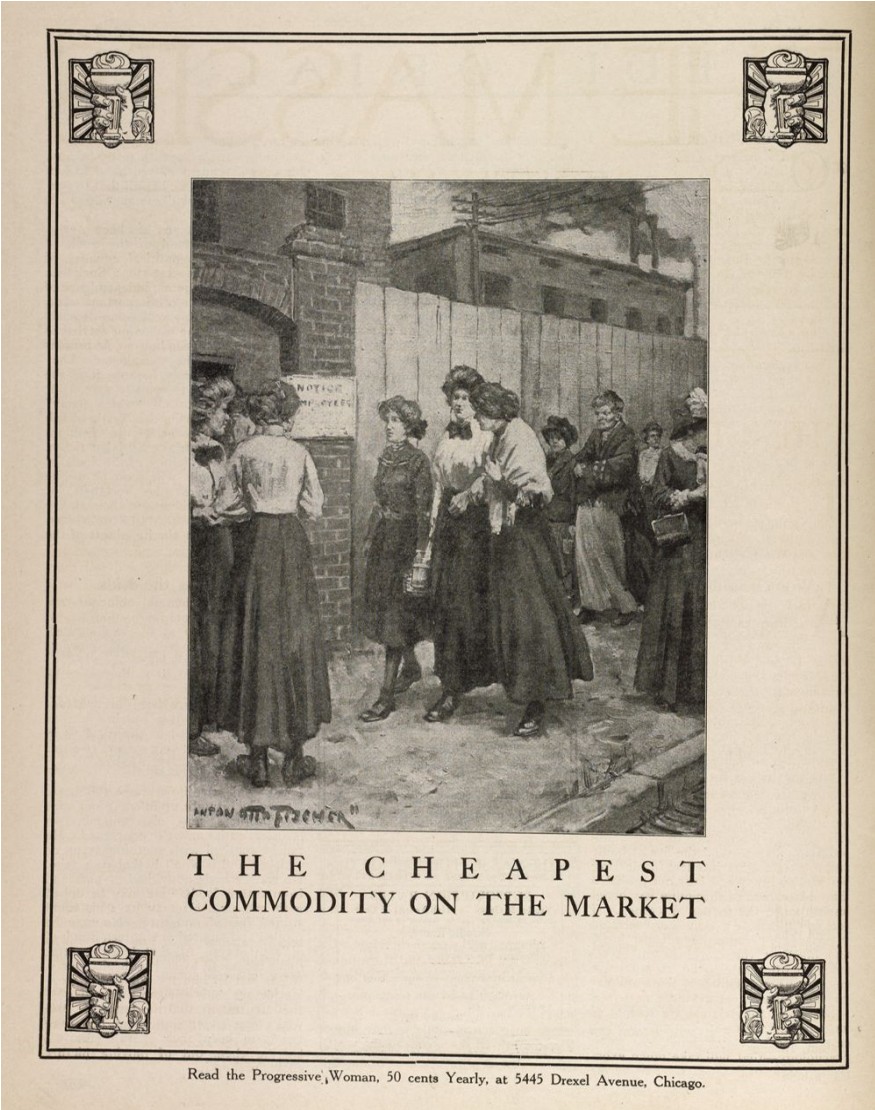

**Figure 5.** The Cheapest Commodity, frontispiece, by A.O. Fisher. *The Masses* V. 1. No. 12, (December 1911): p. 4. The Modernist Journals Project. Available online: https://modjourn.org/issue/bdr526900/ (accessed on 29 August 2022).

While seeking to promote better conditions for women and pointing out terrible injustice, this press example also participates in circulating anxieties prompted by the rapid rise of women in the American workforce. From fears of women taking men's jobs as cheaper labor to protests against women's work outside of the home as "unnatural", with social forces that included a rising swell of immigrant arrivals and animated voices advocating women's rights and other feminist causes, much of the American public confronted the gendered dimensions of capitalism's rise with ambivalence, confusion, and resistance, shaped to different degrees by class, ethnic, and race-based contexts. Visual tropes attached to wage-earning women, as in this example from *The Masses*, suggest widespread tendencies across print culture to align women workers with standard notions of femininity, visualizing types of women's paid work to echo labor undertaken in the domestic space (such as shop girls or weavers), or depicting female victimhood within the labor market. Women, and men, certainly suffered as victims of inhumane industrial labor conditions, most notoriously exposed by the Triangle Shirt Waist Factory fire of 1911, in which 146 workers locked

in the factory were killed when a rapid fire spread through the building, most of them young Jewish and Italian immigrant women (Figure 6). The deadliest industrial accident in America's history at that point, the fire occurred blocks from Ridge's home. However, as demonstrated by the strikes and protests for better working conditions that arose before and in response to this event, including protests conducted by the Triangle workers prior to the fire, women laborers sought representation as fighters against capitalistic abuse, railing against the victimization and dehumanization of all workers.

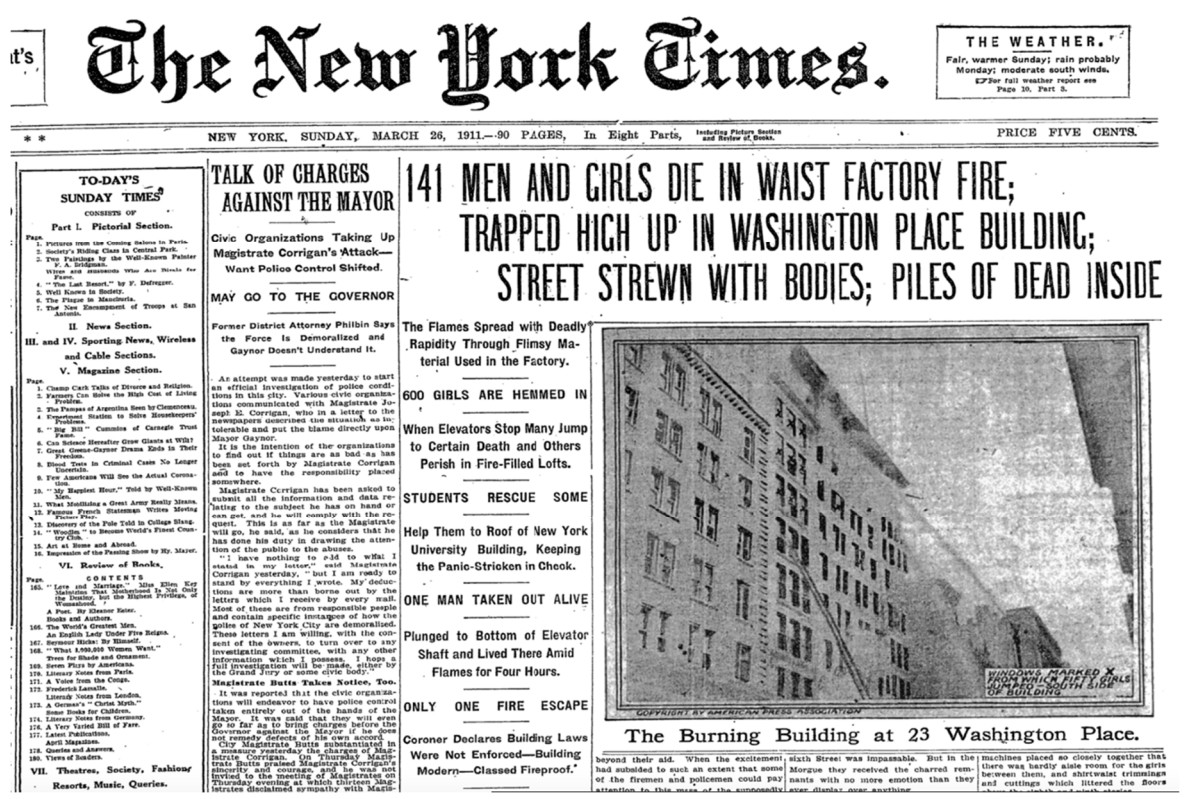

**Figure 6.** Triangle Shirtwaist Factory fire coverage in *New York Times*, 26 March 1911.

"Bread" does not deny, and indeed emphasizes, the mechanisms of capitalism that prey upon the poor, but the poem's argument refuses acquiescent victimization through collectively voicing human need as a priority for economic and social justice. While the women of "Bread" are not depicted as workers within the economic system in the conventional sense of wage-earning labor, they present an image of women's participation in the economic system from spheres usually overlooked in market models, and their presence makes visible the particularly gendered labor that capitalist markets elide. The collectivity of female bodies demonstrating for a non-profit-oriented system of economic justice—a system in which meeting human needs is primary—intervenes in the masculine vision of "work" or "labor" and the capitalist model of economic man, the autonomous market agent. Ridge's poem suggests a labor unrecognized by the models generated by modern economic theory and sustained by capitalist systems, particularly as this labor is associated with women, with women's bodies, and with human need.

In "The Ghetto", the "enduring flesh" of women and their progeny echoes this collectivity. Moreover, it visualizes mothers as the vanguard for change, pressing through generations of care and endurance to become masses that "surge" through the streets. The poem's opening section presents the urban market thoroughfare of Hester Street crowded with close-pressed bodies, associatively shifting to images of an ancient past of Jewish mothers in Egyptian slavery:

> Brood of those ancient mothers who saw the dawn break over Egypt . . .
> And turned their cakes upon the dry hot stones
> And went on
> Till the gold of the Egyptians fell down off their arms . . .
> Fasting and athirst . . .
> And yet on . . .
>
> Did they vision—with those eyes darkly clear,
> That looked the sun in the face and were not blinded –
> Across the centuries
> The march of their enduring flesh?
> Did they hear –
> Under the molten silence
> Of the desert like a stopped wheel –
> (And the scorpions tick-ticking on the sand . . . )
> The infinite procession of those feet? (Ridge 1918, p. 5)

The final stanzas repeat the word "flesh" four times, transfiguring the bodily crowdedness of the street into an embodied, inherited, and specifically feminized vision of collective force and endurance. The word "flesh" signifies both the oppressed body and the body of resistance in a stanza that moves from a wide-angle view of the street to a focus on the "ancient mothers", envisioning a "march" and "procession" across generations, extending a legacy from ancient times into the modern moment (Ridge 1918, p. 4). Juxtaposing imagery of mass marches, young women, and generations of mothers, the poem evokes a legacy for the stunning sight of women leading protests and marches to demand equal rights, wages, and working conditions.

This opening gambit, envisioning masses marching through history and on the modern street, appealed to at least one early reviewer: Ridge's fellow New York poet Alfred Kreymborg. His review, appearing in *Poetry* magazine soon after the book's publication and suggesting how contemporary readers might react to its unconventional poetics wedded to socio-political concerns, explicitly locates the title poem's modern experimentation in its visual emphasis and its connection to an economic and social critique of American capitalism. Gesturing, perhaps, toward Ridge's earlier training as a visual artist, the review calls her a "revolutionist" who is "first and always an artist", and predicts that the poem's powerfully visible attention to economic injustice would unsettle the "average American gentlefolk who are so content with conditions as they are that they never disturb themselves as to their composition or de-composition" (Kreymborg 1918, pp. 335–56). Kreymborg specifies the poem's "deadly accuracy of versatile *images*" and "*portraiture* of emotion and thought", praising the poem's revolutionary "excoriation", and citing the title poem for building a "realistic presentation" of the East Side through particularly visual means that highlights the area's modern ethnic diversity, noting the "*magnificent pageant* of the Jewish race" illuminated in the poem (Kreymborg 1918, p. 337, emphasis added).

Kreymborg's language of a *collective* visibility, echoing Ridge's march of women and their progeny through history, suggests not only suffrage marches and labor strikes but also their spectacle and theatricality. Women's "embodied, gendered presence in physical public spaces traditionally reserved for men", the "colorful spectacle of thousands of American women marching in formation down Fifth Avenue", with "suffrage banners and parades" were all "significant elements of modernism" on the streets of Kreymborg and Ridge's city (Chapman 2014, pp. 13–14). Just as immediate, the Paterson Strike Pageant, organized in 1913 by John Reed and other Greenwich Village artists and writers, such as John Sloan and Mabel Dodge, and the Ferrer School sought to dramatize the ongoing Paterson silk strike.[8] The pageantry, involving 2000 actual strikers, was held in Madison Square Garden and was meant to educate the public.[9] Reviewing it, *Survey* magazine testified to the persuasive power of visual images and representation in observing that the "average man who went to look [at the Paterson strike pageant] and the social observer familiar with

labor struggles left Madison Square Garden with a vivid new sense of the reality of the silk strike and of industrial conflict in general for that matter" (in Kornbluh 1964, pp. 210–14). *Survey*'s language of a "vivid . . . reality" conveyed by visual spectacle richly underscores Kreymborg's comments on pageantry in a poem opening with images of "ancient mothers", envisioning a "march" and "procession" across generations. Ironically, but underscoring the tendency to visually represent workers as masculine, the IWW poster advertising the Paterson Strike Pageant features a male figure clamoring over industrial buildings to burst from the page in defiance (Figure 7).

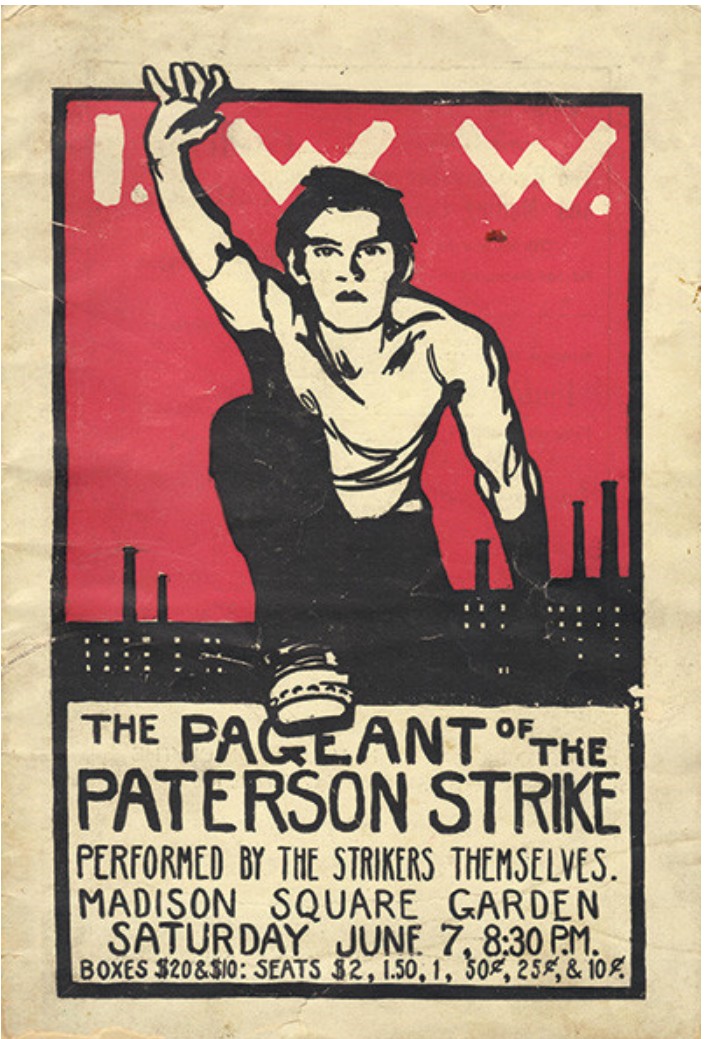

**Figure 7.** IWW poster for Paterson Strike Pageant, 1913. Designed by Robert Edward Jones. Wikicommons.

Alternatively, factory labor in "The Ghetto" is represented through female figures of a younger generation. In the second section, focused on a trio of working women, the speaker adopts the female voice of a Lower East Side resident (as was Ridge), introducing other women living in her tenement who hail from Jewish immigrant groups and work in the garment industry. While the poem highlights individualized portraits of three such women (Sadie, Sarah, and Anna), they are contextualized by a broader and more collective portrait of women's solidarity and agency in the urban ghetto. Within the lineage of strong masses of women, the modern "new woman" is figured in the poem as an industrial worker and political agent, joining with other women. They move through the streets freely in groups, their "heads are uncovered to the stars,/And they call to the young men and to one another/With a free comraderie" (Ridge 1918, p. 4). The speaker is careful to note that

women appear at the raucous "committee" meeting of laborers and socialists (depicted in Section 7), as among the "Words, words, words" and the "Egos" of men crowding the room, "Here and there a woman . . . " (Ridge 1918, p. 19, ellipses in text). This small detail, and the ellipses that follow as though opening up on unspoken possibilities for women's voices, suggests the strategic importance women recognized in preparing for participation in labor and political organizing. The WTUL's *Life and Labor* magazine, for example, included a series written by Margaret Dreier Robins, appearing serially across 1911, on "How to Take Part in Meetings", instructing women in parliamentary rules, structuring and chairing meetings, and engaging in discussions. Ridge's images of the meeting, particularly through unflattering images of young men talking over one another and competing, suggest the benefit of women's increased participation at such gatherings and in public spaces. For Ridge's young women, the marvel of moving through and laying claim to public spaces signifies a particularly modern phenomenon, figured in media images of crowds of women workers taking over the street, whether it be closing time for shopgirls (Figure 8) or a labor parade (Figure 9).

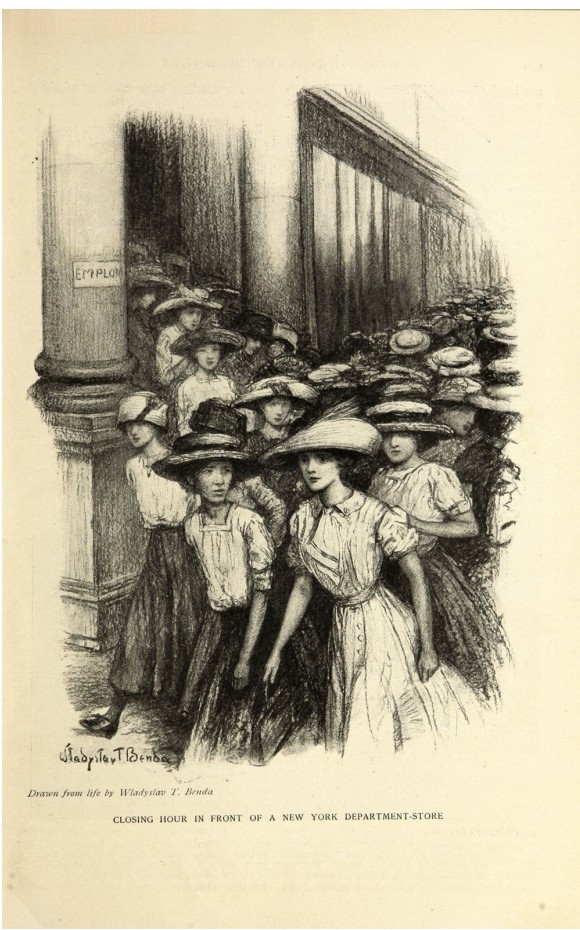

**Figure 8.** Closing hour at a New York Department Store. *McClure's Magazine* 35.6, Oct. 1910, p. 599. The Modernist Journals Project. Available online: https://modjourn.org/issue/bdr554781/ (accessed on 29 August 2022).

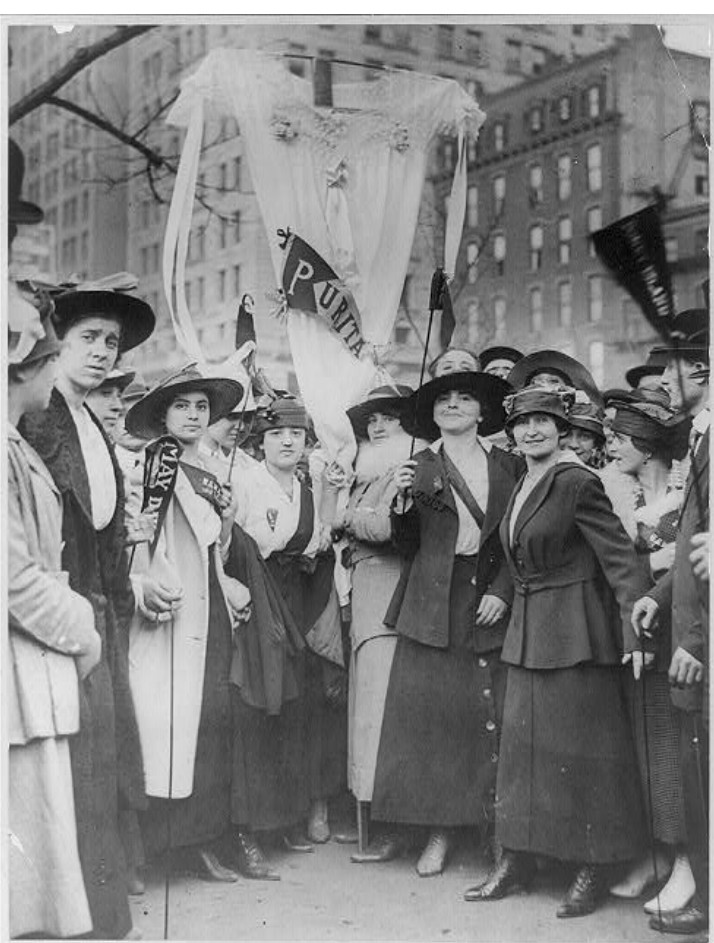

**Figure 9.** Bain News Services. Garment workers parading on May Day, 1916, New York. Library of Congress.

### 3. Documenting Women at Work

"The Ghetto" focuses on the image of the woman laborer most directly and extensively in its second section. Following the first section's "pageantry" of women on the streets, the poem introduces three garment factory workers—Sadie, Sarah, and Anna—who live in the same tenement as the speaker. Sadie's factory work takes center stage in a dramatic manner, alongside details of the women's personal, intellectual, political, and social lives. Sadie's portrait most fully represents the physical demands of her workplace as she combats the industrial sewing machinery. Together, this cluster of poetic portraits militates against the visual prominence of male workers in much print culture, while also engaging with social conventions and anxieties attached to female wage-earners. As socio-visual contexts for discussing these poetic portraits, the final two sections of this essay discuss several distinct segments of print culture that visually incorporated the industrial working woman, to different purposes: post-bellum periodicals; women's labor publications; social documentary photography by Jacob Riis, Lewis Hine, and Frances Benjamin Johnston; and illustrations in *The Masses* by John Sloan.

By 1870, when "over one-third of New York City's workforce was female" (Balliet 2007, np), cultural anxiety over the social effects of women working coincided with ambivalence over how to represent the woman worker and the range of work she undertook across different classes, and in response to the nineteenth century's relocation of women's work from home to an industrial setting. Historians looking at the rise of the illustrated weeklies point to post-Civil War print images of working women that present either cautionary tales about the social dangers of women working (i.e., prostitution, abandoned children, etc.) or idealized depictions of work that fit acceptable codes of femininity. Barbara J. Balliett points

to an 1868 *Harper's Bazar* two-page spread, accompanying an article on the progress women had made in previous decades. Entitled "Women and Their Work in the Metropolis", the image presents women working singularly or in small groups, often in domestic settings, and centered with the large image of a woman in a rocking chair, sewing beside a cradle (Figure 10). The smaller images surrounding her depict new industrial opportunities for women, such as box making, fur sewing, candy sorting, silver burnishing, or shoe fitting. Women are neatly dressed and often in what seem to be convivial groups, within environments that suggest light-filled, comfortable interior spaces. Although women are featured with machines in images of type-setters and collar makers, the dominant sense is of a domestically replicated factory or work environment, reinforced by the central image of the seamstress performing hand-work beside the baby's cradle (Balliet 2007, np). An earlier image, depicting women's increasing entry into factory labor, appears in *Ballou's Pictorial* and depicts women at work in the paper mills with the new "Foudrinier" machine used for making paper; such images often positively underscored the potential in factory work for women's self-sufficiency, independence, and savings capacity (Figure 11).

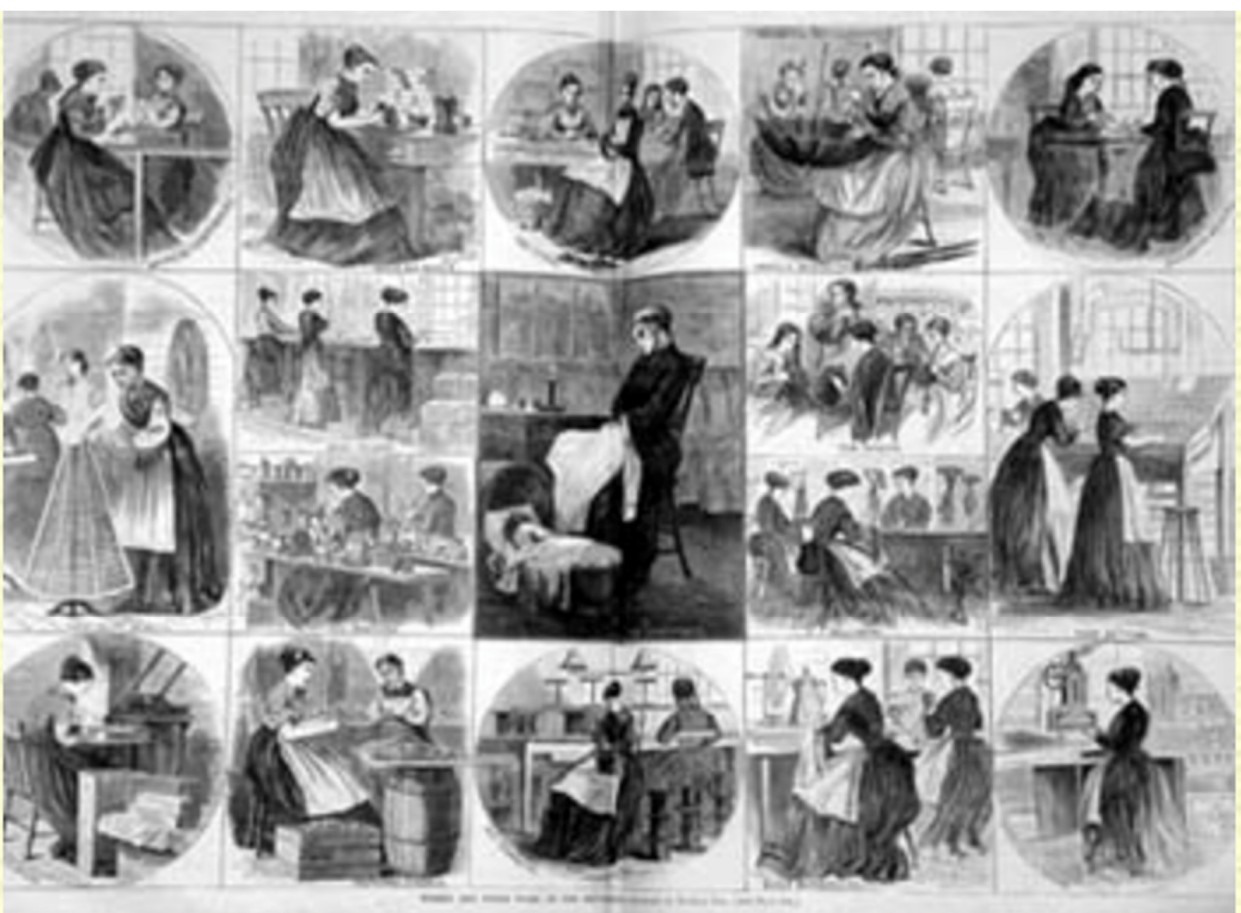

**Figure 10.** "Women and their Work in the Metropolis." *Harper's Bazar* (April 1868). The American Antiquarian Society. Available online: http://commonplace.online/article/let-them-study-as-men-and-work-as-women/ (accessed on 30 May 2022).

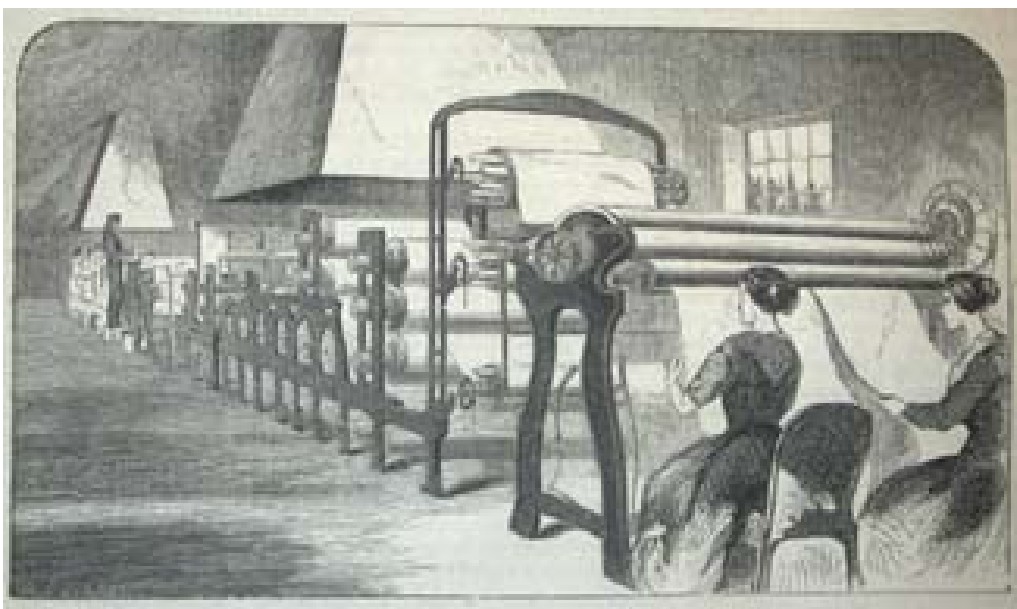

**Figure 11.** Women at Foudrinier machine. *Ballou's Pictorial*, June 0, 1855. American Antiquarian Society. Available online: https://www.americanantiquarian.org/Exhibitions/Womanswork/factory.htm (accessed on 29 August 2022).

By the turn of the century, labor tensions encouraged less positive or mollifying views of factory and industrial conditions; nonetheless, depictions of modern labor persistently drew upon culturally sanctioned notions of femininity and masculinity. As will be discussed, Ridge's portrayal of Sadie captures her skill with machinery amidst poor work conditions for industrial women, departing on both counts from post-bellum imagery reassuring wage work as properly feminine. Ridge's picture of the garment factory supplies a visual image of women typically overlooked (or willfully ignored) in much coverage of labor, even in venues sympathetic to the labor cause. The preponderance of images of women laborers that did circulate showed them posing for portraits or in the act of striking or engaging in domestic work but rarely as women actually working in industrial settings or in high-skilled jobs.[10] In her study of *The Masses*, Rachel Schreiber makes the important point that it is not merely that images of "women actually working are rare in the radical press" but that when they do appear, they are presented as "victims of unfair labor practices .... [and] passive symbols of the woes of working-class life and the evils of the sweated labor system" (Schreiber 2011, pp. 52–58).

Print culture more commonly displayed the modern worker as male, and, even as increasingly large numbers of women entered wage labor, the masculine image of the laborer persisted from the nineteenth into the twentieth century. As Nan Enstad and other labor historians specify, " ... the *ideal* of 'worker' in the nineteenth century was male" (Enstad 1999, p. 3). He was also white, Enstad emphasizes, referencing David Roediger's observations that "male labor unions in the antebellum era developed a heroic category of 'worker' for white males, in direct contrast to 'slave labor.' Their language of class formed around concepts of 'manly' and 'free' labor that explicitly excluded African American laborers and all women" (in Enstad 1999, p. 3). Extending into the twentieth century, the image of the heroic, viral, skilled, and white laborer fueled a visual rhetoric that largely "assumed the gender of the 'worker' to be male" (Schreiber 2011, p. 60). The pages of *The Masses*, for example, a major supporter of labor, promoted visual representations—drawings, cartoons, photographs—that asserted the "ideal male worker as a powerful force who could address the wrongs of industrial capitalism", an image gaining renewed force during economically challenging years (Schreiber 61). By the time of the Depression, the popularity of publications such as Lewis Hine's 1932 *Men at Work: Photographic Studies of Men and Machines* joined heroically masculinized figures in public murals, FSA photographs,

and other visual records to celebrate skilled labor as emphatically masculine. Hine's earlier documentary work, more contemporary to Ridge's labor poetry (and to be discussed shortly), often focused on women and children, but the figure of the muscular man in tandem with modern machinery caught his camera's attention long before the 1932 book. His 1920 image of the "powerhouse mechanic" exemplifies the camera's insistence upon the man as co-equal with the machine, compositionally echoing the machine's rivets in the sculpted and defined muscles of the mechanic, suggesting a symbiosis of man and machine (Figure 12).

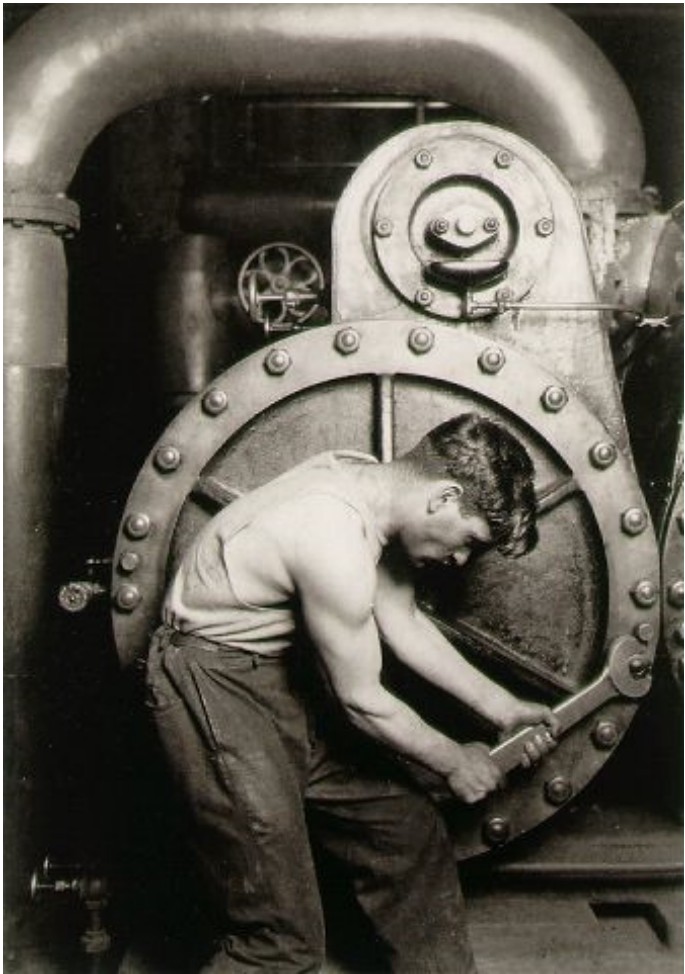

**Figure 12.** Lewis Hine 1920. "Machine Powerhouse Mechanic Working on Steampump." Wikipedia Commons.

To find imagery of women at work, we might turn to the women's labor movement and efforts to gain greater recognition and visibility. Ridge's focus on women and labor resonates deeply with the activism and ideological arguments of *industrial feminism*, named so in 1915 "by scholar Mildred Moore to describe working women's militancy" since 1906 (Orleck 1995, p. 54). During the early years of the labor movement, women's exclusion from men's unions led industrial feminists to organize on behalf of working-class women's issues. Industrial feminists worked in co-operation but also in conflict—especially concerning class issues—with other advocates for women, particularly a less class-conscious suffrage movement.[11] Denoting a political consciousness galvanized around a "vision of change", these working-class feminists asserted issues of human need—wages, working conditions, child and health care, education—as central to economic understanding and policy within an intensifying capitalist industrialism (Orleck 1995, p. 54).

Trade magazines dedicated to women's labor tended to picture working-class women as activists more so than as workers engaged in labor. The *International Ladies Garment Workers' Union* (*ILGW*) magazine exemplifies the visual emphasis on activism more so than on labor itself. For example, running from 1910 to 1914, the journal introduced readers to engraved or photographic portraits of "young women like seventeen-year-old Minnie Labetsky, whose employer offered to bring her mother to America if she would cross the picket line. 'I would rather die than go back to work, to scab,' Minnie reportedly vowed", her activist's pledge accompanied by a "smiling photo" of her (Orleck 1995, p. 78). In a well-known portrait, Rose Schneiderman, a vocal advocate for working women and later head of the New York State Department of Labor under FDR, sits smiling at her sewing machine, as though posed running a piece of fabric through a small sewing machine (Figure 13). Such positive imagery might well have been calculated, in part, to dissuade native-born characterizations of "Jewish and Italian immigrants" as "unhealthy and racially degraded."[12]

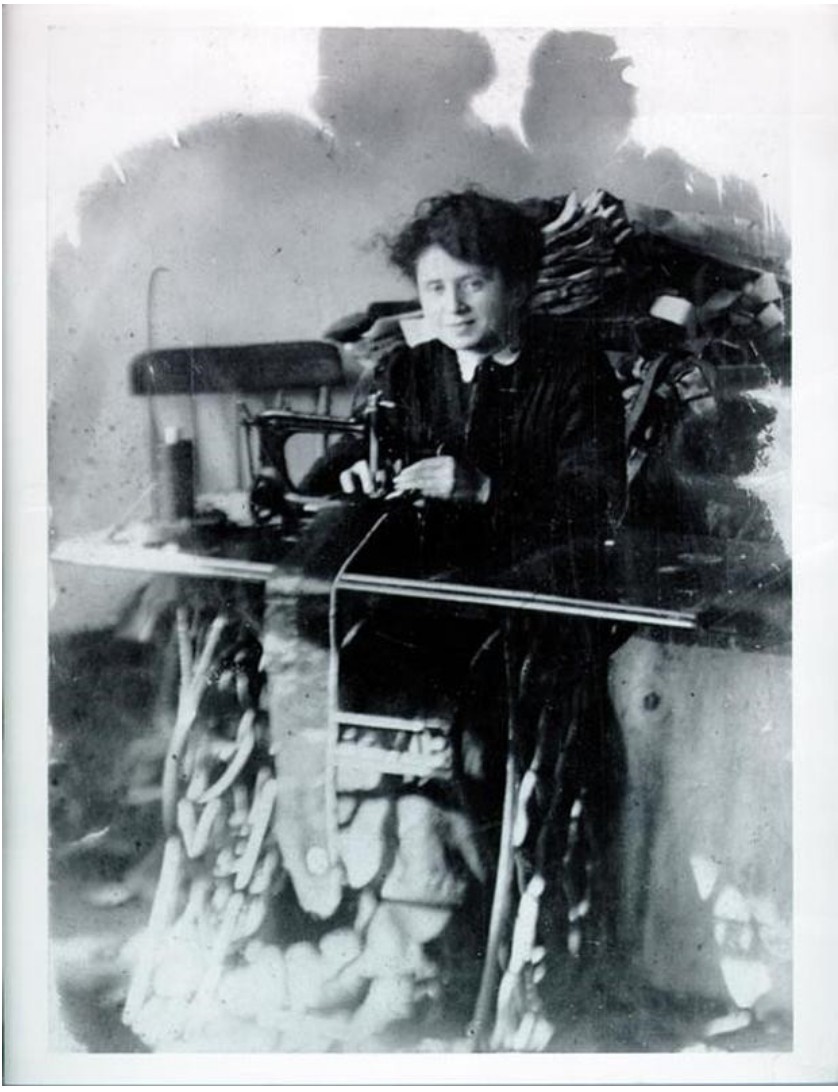

**Figure 13.** Rose Schneiderman sews a strip of fabric on a machine. Image courtesy of Kheel Center for Labor-Management.

The visual presentation of women activists underscored a drive to proclaim the importance of women to industry and positive reform. Arguing for greater self- and public recognition of the industrial woman worker in the fight for the vote, the labor activist Pauline Newman declared in *The New York Call*: "At this time, [woman] is first beginning to

wake up to the fact that *she is an industrial factor in society*, and is, as a consequence, taking her place in the labor movement, when she is beginning to realize her economic power, she will ... use the ballot to back up that economic power ... and, slowly but surely achieve the end—economic freedom."[13] Newman's insistence, like the *ILGW* and Ridge's garment workers, stresses women's significant participation in the labor force by the 1910s as a corrective to a "paucity of images of women workers" in the mainstream and radical press (Schreiber 2011, p. 51). By 1911, the National Women's Trade Union League (WTUL) began incorporating a more diverse range of visuals into its new magazine, *Life and Labor*. Running for several years as an organ for the league, the monthly periodical combined essays, opinion pieces, announcements, poetry, fiction, photography, and drawings that addressed the range of working-class women's vocations, which included industrial, mercantile, and domestic labors.

Visually lively, *Life and Labor* included journalistic photographs of protests and strikes, alongside portraits of working women, new immigrants, and labor supporters. Visual materials are routinely enlisted to emphasize the magazine's goals of expressing "the forces both latent and active in the woman movement of this country and thus bring the working girl into fuller and larger relationship with life on all sides."[14] The editors, who included Alice Henry, S.M. Franklin, Frances Squire Potter, and Harriet Reid, strove to present a positive picture of the "working girl", especially to mitigate stereotypical images of newly immigrant women as unclean, un-American, and unassimilable, and to stress the importance of immigrants to both the women's and labor movements. One striking photograph, the frontispiece to the February 1913 issue, shows a young woman striker proudly standing in front of an American flag (Figure 14). The photograph's sub-caption suggests her immigrant experience, seeking an American way of life that has failed her and led her to participate in labor activism by joining the White Goods Strike in New York Ciy: "This girl was arrested for trying to obtain the conditions of life she believed were represented by the American flag." Labor leader Rose Schneidermann's report on the strike, "The White Goods Workers of New York: Their Struggle for Human Conditions", follows the photograph, claiming that 7000 workers took part in the five-week January walk-out of workers manufacturing underclothing. Her article includes several individual portraits of women strikers, somber and proud in affect.[15] The frontispiece, encoding the woman's proud pose with patriotic (while ironic) stature, crops her dark-haired head and torso against the backdrop of a flying flag. The photograph's attribution to *The New York World* newspaper, which at the time reached daily circulations of up to one million, suggests a wider circulation of the image's pro-immigrant, pro-labor message.

*Life and Labor* routinely incorporated material from social studies of labor undertaken in the 1900s and 1910s that had begun incorporating photography, developing into a new genre of social documentary. Such photographs proved essential to reports such as the Committee on Women's Work of the Russell Sage Foundation, a series of studies on women's work conditions in New York trades. Reprinted in *Life and Labor*, images by Lewis Hine from the report appear in issues during 1913, evidencing poor, crowded, and unhealthy industrial conditions (Figures 15 and 16). With the advance of photographic and printing technologies in the early years of the century, Hine helped make the camera essential to modern forms of social research, as will be discussed. The rise of social science disciplines and publications in which such visual documents appeared, such as *Survey* (and its later iteration as *Survey Graphic*) aided the Progressive Era's reform effort. The photograph's utility for social work influenced a modern genre of documentation, giving rise to visual vocabularies and methods of documentation reshaping public perceptions of modern labor, albeit shaded by gender conventions. The camera's framing and representation of the woman worker, while seemingly objective, is nonetheless shaped by and carries social attitudes in its visual vocabulary and iconographic layers of meaning.

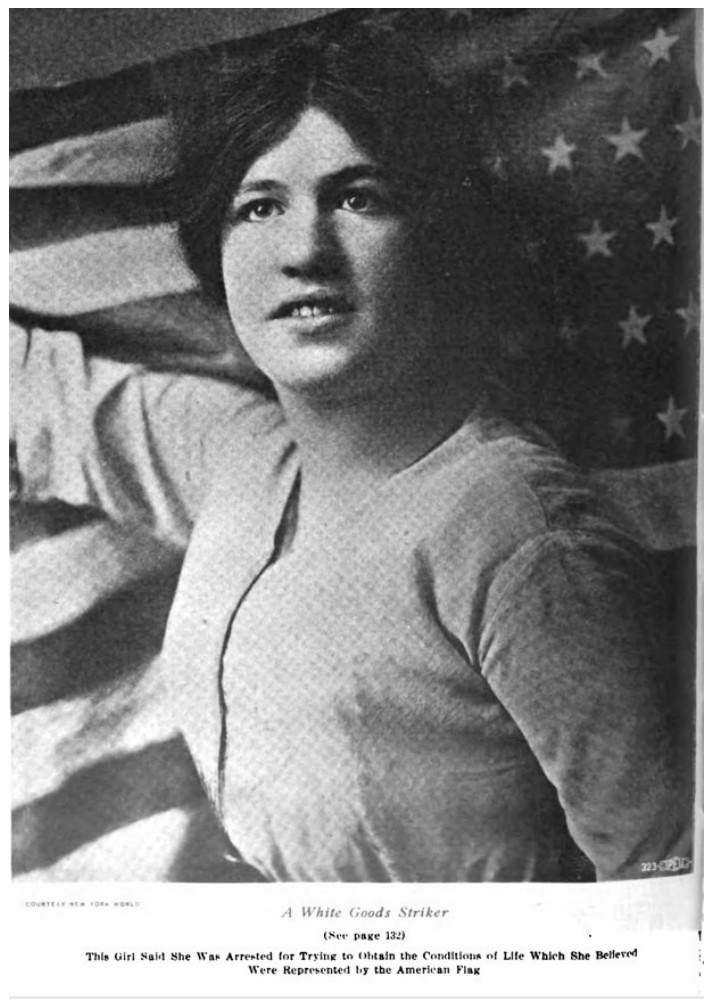

**Figure 14.** *Life and Labor* 1913 (February). "A White Goods Worker." Frontispiece.

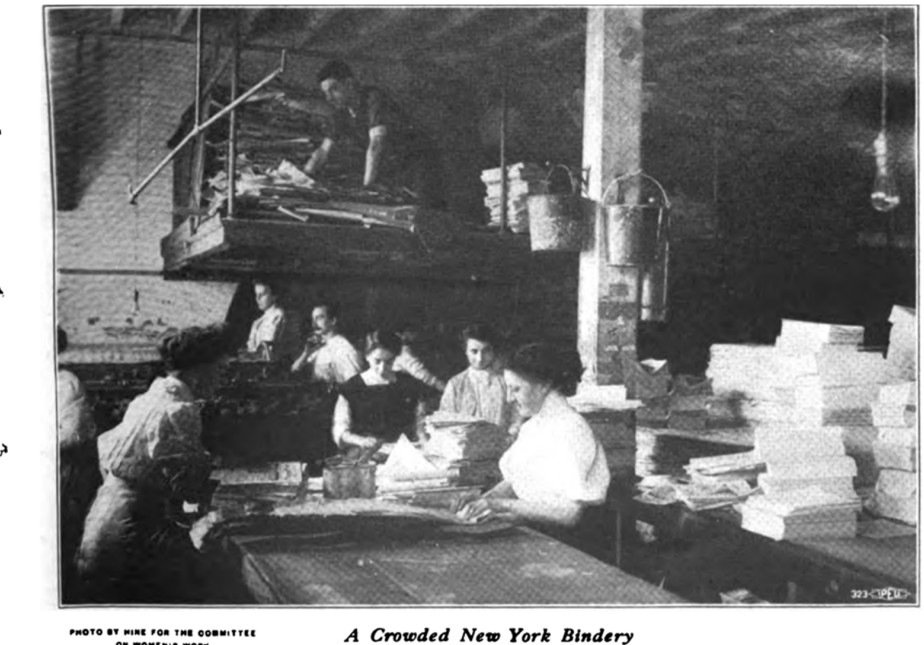

**Figure 15.** Lewis Hine. 1913. "A Crowded New York Bindery." In *Life and Labor* (October), taken from the Russel Sage Foundation Committee on Women's Work.

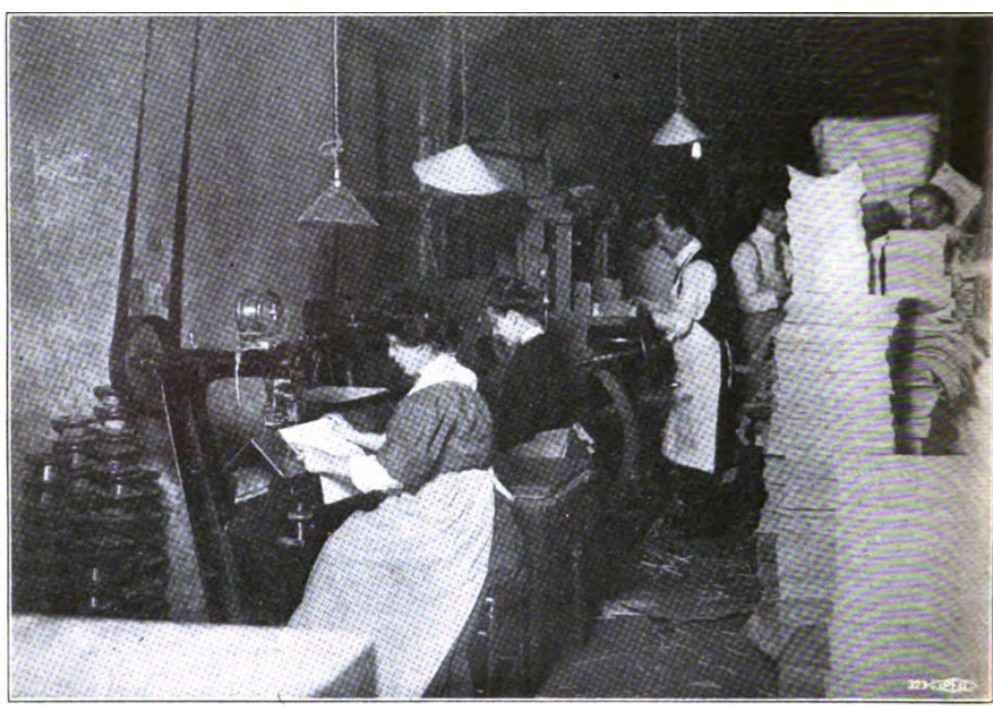

PHOTO BY HINE FOR THE COMMITTEE ON WOMEN'S WORK

*Women Stitchers and Man Smashing in Dark, Unventilated Room in New York*

**Figure 16.** Lewis Hine. 1913. "Women Stitchers and Man Smashing in Dark, Unventilated Room in New York." In *Life and Labor* (October), taken from the Russel Sage Foundation Committee on Women's Work.

Jacob Riis, capturing the public eye through his investigative journalism as a police reporter, was among the earliest to introduce photographic images as a reporting method for social reform purposes. Covering different areas and ethnic/racial groups in New York City, his intimate depictions of crowded tenement homes used for home-piece work and dark, cramped sweatshop conditions gained currency, first through his 1888 illustrated lectures, "The Other Half: How It Lives and Dies in New York." Presenting at meetings of churches, civic organizations, reform groups, and camera clubs throughout the city, Riis used lantern slides of his photos to illustrate the lectures, drawing upon the power of visual media to make his reform-minded argument. Hired to write an illustrated article for *Scribner's* in 1889, he expanded his lectures and images into his 1890 book, *How the Other Half Lives*, popularly selling eleven editions in five years. Women were part of the depicted labor force, most often in family-group and home-piece environments. Riis' images focused on the conjunction of living and working spaces in unhealthy, crowded, and substandard tenements, joining arguments for better housing that viewed poverty as the result of environmental forces rather than of individual weakness while also playing on consumer fears of contagion by immigrant workers in their homes. Particularly by representing women as mothers, often among their children and infants in homes blurring boundaries with workplaces—with piles of piecework or rags cluttering home space—Riis' images elicited outrage and sympathy over the affront to conventional motherhood brought on by the conditions of modern capital (Figure 17).[16]

As factory work increasingly displaced home-piece work for women in the 1900s, and as reporting on industrial conditions occupied social investigators and reformers, Lewis Hine was popularly enlisted to visually document social science studies of labor, poverty, and American industry. Trained in social work at Columbia University, Hine had suggested the idea of photography as a tool of social welfare to Paul Kellogg, the assistant editor of *Charities and the Commons*, a nationally distributed social welfare magazine that soon renamed itself as *Survey*, in which Hine published a series of documentary photographs

of immigrants at Ellis Island. In 1907, Kellogg enlisted Hine to photograph laborers for *The Pittsburgh Survey*, a six volume study edited by Kellog as "the first attempt in the United States to study life and labor in one industrial city in a comprehensive and systematic manner "(Greenwald 1984, p. vii). The extensive six-volume study, which "viewed Pittsburgh as a microcosm of American urban life under industrial capitalism", was widely published in excerpts for *Charities and Commons*, popular magazines including *Collier's*, and in book form (Greenwald 1984, p. viii). As "principle staff photographer" for all six volumes (one of which included *Women and the Trades*), and creating what Kellogg later referenced as Hine's "work-portraits", Hine categorized his work as "social documentary" (in Stange 1989, pp. 47, 51, 53). As Maren Stange argues, studies such as *The Pittsburgh Survey* and Hine's work-portraits foregrounded the laborer with new emphasis: "for the first time in our social science literature, the industrial worker emerged as a distinct American figure presented without quibble or evasion, whose presence was to be acknowledged fully and reckoned with" (Stange 1989, p. 55).[17]

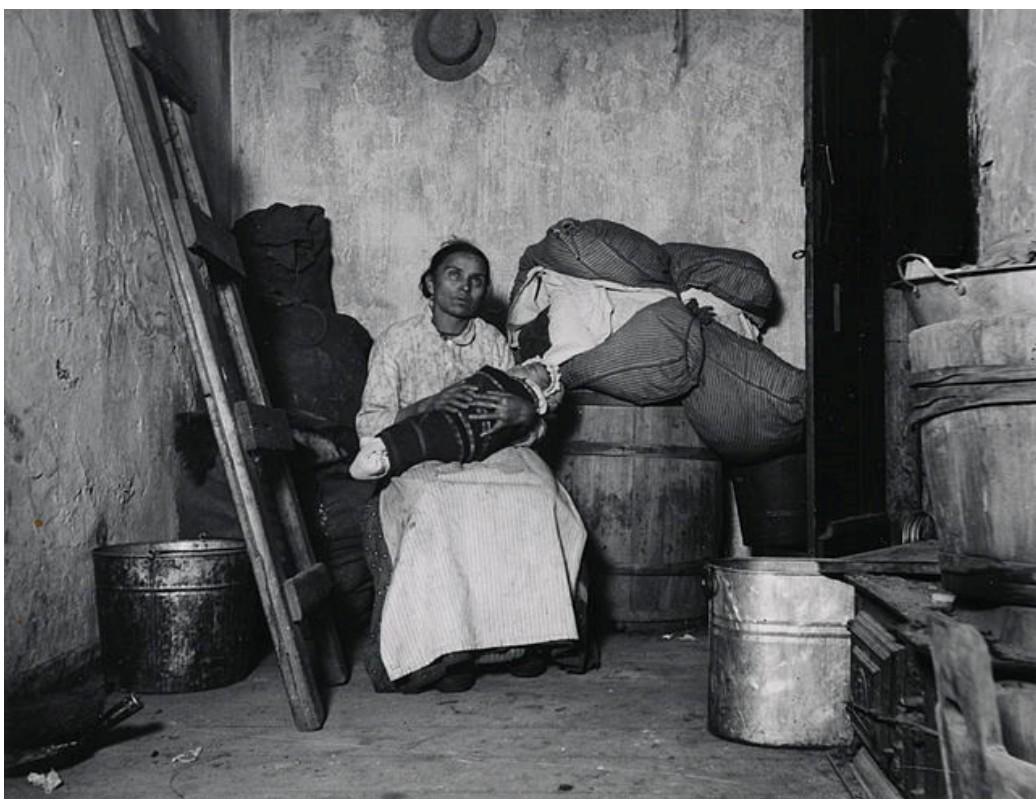

**Figure 17.** Jacob Riis, Italian Rag Picker in Her Home. Wikicommons.

The picture of the "industrial worker" that Hine and *The Pittsburgh Survey* captured included women. In the survey's *Women and the Trades*, author and urban reformer Elizabeth Beardsley Butler concluded that "the fruits of economic growth in the early twentieth century were unequally distributed among wage earners on the basis of gender, skill, and ethnicity" (Greenwald 1984, p. ix). As with his photographs for the Committee on Working Women in New York several years later, Hine focused on Pittsburgh's working conditions, picturing women in industrial and other work environments. Introducing the 1984 reprint edition of *Women and the Trades*, Maurine Weiner Greenwald delineates Hine's documentation of "the rigidity of work processes . . . [and the] highly refined system of supervision" of female workers by male supervisors, and the "routinization of labor" (Greenwald 1984, pp. xxxvii, xl). Commenting upon Hine's visual compositions, she singles out a "full-room photo of female coil workers at Westinghouse, showing long rows of the same machine as far as the eye can see, with each woman seated . . . at her assigned

place" (Greenwald 1984, p. xl). Employing a monumental depth of field and dramatic perspective, this kind of shot served Hine well in photographs he took for the 1908 *Child Labor Report*, positioning small and often female child figures alienated and overwhelmed by the looming machinery (Figure 18).

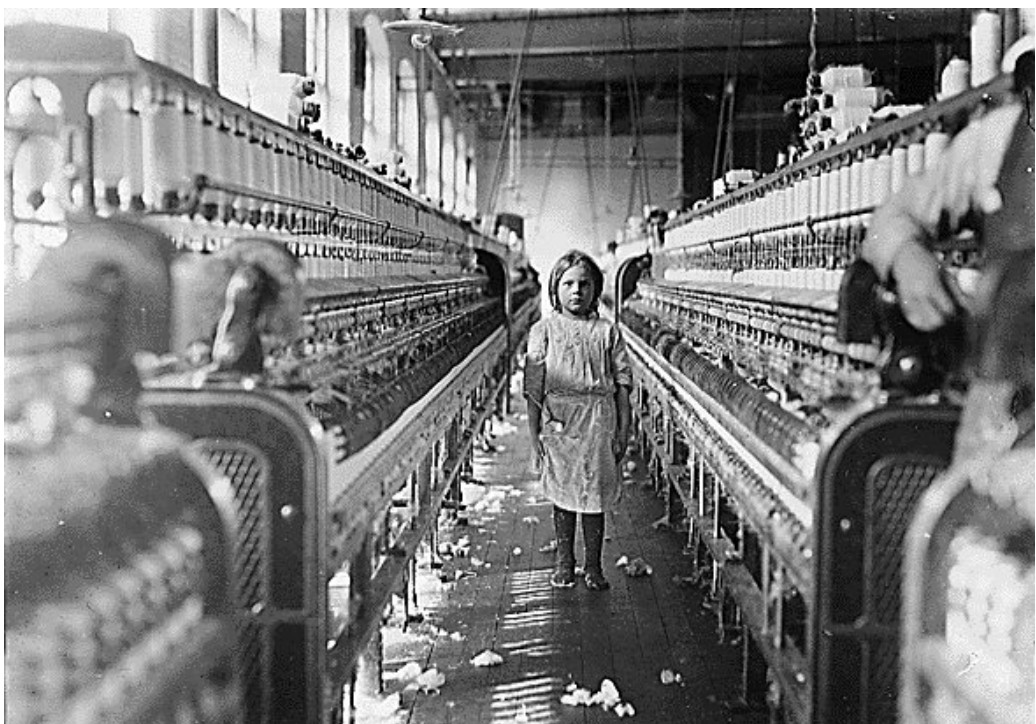

**Figure 18.** Lewis Hine, 1908, New England Mill. Library of Congress.

A distinctively visual rhetoric of laborer and machine crosses from photographic documentation to poetic visualization in "The Ghetto." Consider Sadie's representation: depicted at work with her machine, the first extended view of Sadie stresses visual facts and compositional dynamics of worker and machine similar to evolving uses of photographic evidence in the emerging social-science field studying social problems. In the figure of Sadie, the poem insists upon women's bodies being positioned within industrial and cultural forces. Emphasizing unsafe and inhumane factory conditions, the poem draws the workplace scene unsparingly:

> All day the power machines
> Drone in her ears
> All day the fine dust flies
> Till throats are parched and itch
> And the heat—like a kept corpse—
> Fouls to the last corner. (Ridge 1918, p. 7)

As the poem's most vivid picture of industrialized factory labor, this image of Sadie presents the female body assaulted and strained, exemplifying the "cheap labor driven to work at inhuman speeds under wretched conditions" that the *Pittsburgh Survey's Women and the Trades* volume described as the speed-as-profit system driving the garment industry (Greenwald 1984, p. xxii). Sadie's body is "keyed to the long day" of sewing machines (Ridge 1918, p. 7).

Initially in this section, the poem signals modern industrialism's eviction of creative forms of labor that connect the worker's experiences with the product of her/his labor. This loss is figured through Sadie's father, Old Sodos, a male saddle maker in the old country who, in migrating to America's industrial environs, has "forgotten how" to hand craft

(Ridge 1918, p. 5). In the factory, such craft is also forgotten amidst industrial forces. Sadie's movements are regulated by the repetitive labor that makes her "One with her machine", the "biting steel—that twice/Has nipped her to the bone" (Ridge 1918, p. 6). Using similar language in describing modern developments in the garment industry, the author of the 1909 *Women and the Trades*, Elizabeth Beardsley Butler, recounts entering a garment factory door:

> . . . you are greeted with a roar of wheels and a quivering of the floor that bear witness to the harnessing of the sewing machine. The sewing machine itself we think of as a late development. Many among us can remember when we hailed it as a time saver, a burden bearer of our individual cares. It is just at this point that the further development has come. (Butler [1984] 1909, p. 102)

The "application of steam power to the sewing machine principle" of the "new machine" enables "quick production" of "ready-made garments" in increasing demand (Butler [1984] 1909, pp. 102–3); "the power carries with it its own tenseness. The operators feel the spur and never relax" (Butler [1984] 1909, p. 103). Introducing Butler's chapter on garment workers, Hine's photograph shows women crowded in a dark workroom with piles of fabric around them as they work through each piece, engaged in a repetitive routine. In a similar photograph, taken in a cigar (stodgy)-making factory in Pittsburgh's Hill District, Hine offers a wide frame view of the factory floor that captures dirty, unlit, and crowded conditions (Figure 19).

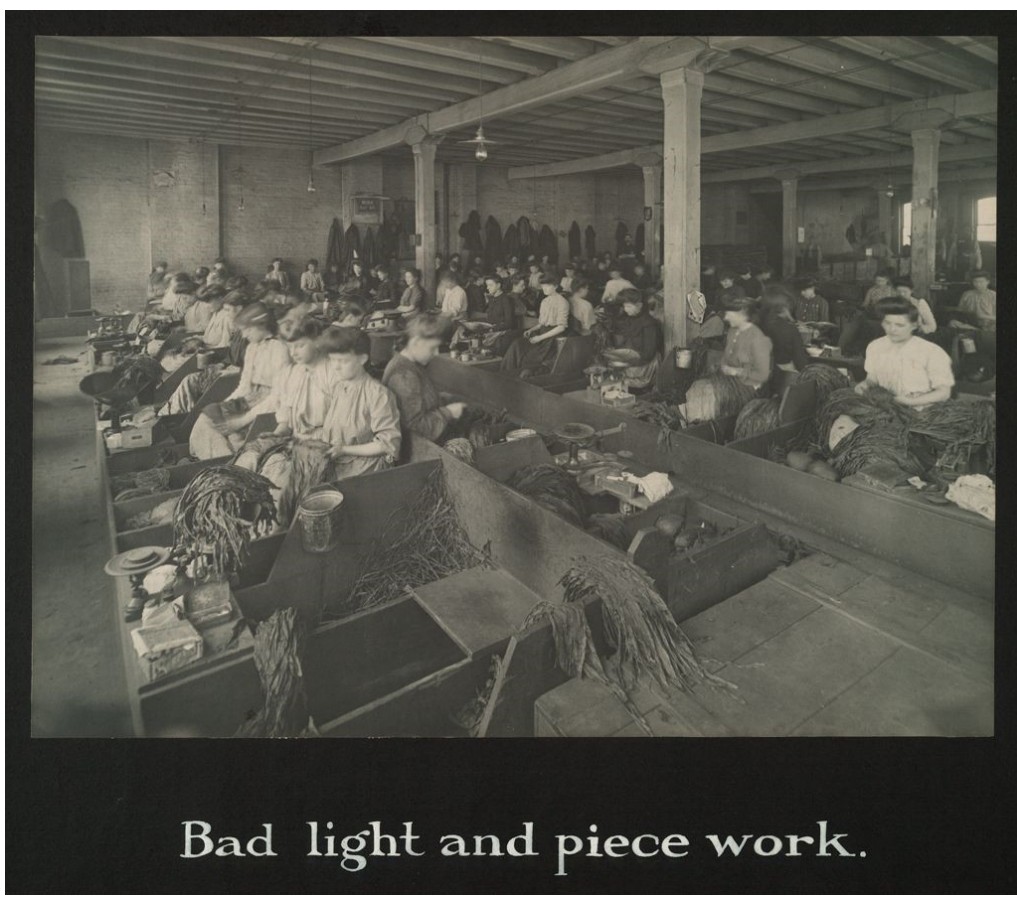

**Figure 19.** Lewis Hine 1909. Stripping Tobacco by the Pound. Image included in Butler's ([1984] 1909) *Women and the Trade* (in Greenwald 1984, p. 78). Image courtesy of Harvard Art Museums/Fogg Museum, Transfer from the Carpenter Center for the Visual Arts, Social Museum Collection.

Like Hine and Butler's depictions, Ridge's factory floor is marred by dust, poor light, and noise. Nonetheless, in this dehumanizing environment, Sadie's relationship with the

industrial sewing machine serves a dual function, showing both its speed-driven toll on the body and the woman's skill in managing it. When the body tires, Sadie fights fatigue to keep up with the machine, driving her body to compete to the point that she is described in gear-like, mechanical terms—a part of the industrial machine. She "quivers like a rod ... /A thin black piston flying", as she "stabs the piece-work with her bitter eye." She is a "fiery static atom,/Held in place by the fierce pressure all about" as she "Speeds up the driven wheels/And biting steel—that twice/Has nipped her to the bone" (Ridge 1918, p. 6). In manufacturing's adaptation of the "science" of modern production, Taylorism's virtues of efficiency, speed, and high productivity render the body's well-being negligible.

Although the worker's relationship with her material is hostile, the imagery also suggests a form of shared energy and a high level of skill. Although Sadie "stabs the piece-work", she is described in language aligning her with the machine that constricts and regulates her. Her simultaneous resistance to and synchronicity with the machine's power suggestively counteracts prevalent images of women's mechanical incompetency. This image of woman and machine rejects cultural conventions associating rationality and technical mastery with masculinity and that "considered technology primarily the province of men and fundamentally foreign to women's ways", underlying stereotypical but "persistent doubts about women's capacities to understand, much less operate, machines" (Wosk 2001, pp. 3, xi). Ridge's attention to women's specific labor in the garment factory, in the figure of Sadie, displaces the privileged iconography of the male laborer, refuting stereotypes of women as passive, weak, irrational, or naturally unequipped to perform physically demanding labor. Sadie's womanhood proves an equal match for the machine, even as the physical struggle and harmful impact on the body are made clear.

The poetic snapshot of Sadie at work registers a skill with machinery that also caught the attention of pioneering woman photographer Frances Benjamin Johnston in several series of women in industrial labor beginning in the 1890s. At a time when sparse visual rhetoric connected women, industry, and work, Johnston's photographic studies of workers explore *how* camera vision positions the iconography of femininity in relation to the iconography of industry and what this positioning suggests about gender and work as constituents of capitalist modernity. Johnston's photography manifests an evolving visual rhetoric of the woman worker that, in the alternative medium of poetry, Ridge's portraits also suggest in the 1910s. In redressing the dominant gendered iconography of the laborer as male, the correspondences between the poet and photographer's respective modes of visually representing the woman factory worker challenge the persistence of naturalized gender norms that located women in the home or in home-like professions, as unable to combat victimization by wage labor, or as unskilled. How might the specular relations of this documentary medium, turned upon women's labor by a woman photographer, suggest ways of reading Ridge's "snapshots" in "The Ghetto"?[18]

Well known in her day but largely underacknowledged now, Johnston studied art in Paris, subsequently setting up a photographic portrait studio in Washington D.C. and attaining great success in art, commercial, and news venues for the new medium. Associated with the Photo-Secession and also supplying images to the Bain News Service syndicate, Johnston photographed high society in D.C., becoming the White House photographer for several presidents, while amassing a photo gallery of notable Americans from Mark Twain to Natalie Barney to Djuna Barnes. She also produced a series of photographs of Black students and their environment at the Hampton Institute, hired by the school to document their educational institution at the turn of the century. By 1913, she and her partner Mattie Edwards Hewitt opened a studio in New York, traveled in bohemian circles, and began extensive documentation of New York architecture and gardens.[19]

Locating her photographic practice in New York in the early teens, Johnston's residence coincided with this intense period of women's labor activism and its visibility in print media and on the street. Surely such woman-centered demands for better wages and industrial work conditions would have galvanized her attention. By the time she arrived in the city, her photographic documentation of women laborers made up a significant part

of her portfolio. A series of photographs, beginning in the 1890s, focused on industrial workers—iron workers, coal workers, New England mill workers—and included several series featuring women in industrial settings and distributed on the early Bain news service. These industry photographs suggestively explore a visual iconography of women and labor that changes over time in relation to differing gender conventions and photographic conventions. Tracing Johnston's work from 1890 to 1910 offers a visual apprehension of the relationship between representations of women laborers and gendered attitudes relevant to Ridge's poem.

Producing some of the earliest photographs of women in industrial forms of labor, Johnston recorded workers at the Washington Bureau of Engraving and Printing, from 1890 to 1895, many of whom were women.[20] An early image, "Woman trimming and stacking currency sheets", calls to mind the popular tendency in the 1890s to emulate painting through the new medium of photography. The camera's treatment of the female subject's dress, hair, and posture, along with elements of lighting and the spatial arrangement of a frontal composition, clearly align the photograph with the pictorialist aesthetic of the time. Published in an illustrated article entitled "Uncle Sam's Money" in *Demorest's Family Magazine* (1890), the image of a woman sitting alone before a desk stacked with sheets employs pictorialism's gauzy, almost painterly approach, tendencies that also stress the woman's femininity (Figure 20). The space surrounding the woman is enclosed but uncrowded, with big windows and architectural details suggestive of domestic rather than industrial space. As though Johnston's experience in the industrial setting altered her vision, a second photograph from this series foregoes the pictorialist aesthetic and its feminizing conventions. In this image, women line up methodically, parallel in the composition to the overhanging industrial lights, as though emphasizing the standardization of work and the workers as they stand to operate complex machinery (Figure 21).

An increasingly modern and industrial visual rhetoric emerges in an 1895 series of photographs at the Stamp Division of the Bureau of Engraving and Printing. Documenting women actively at work and operating machines, these images contest dominant conceptions of women as unsuited for highly skilled labor, as does Ridge's later portrait of Sadie. The images record such activities as "Workers taking mucilaged sheets of postage stamps from the drying box in the gumming and drying room"[21], "men and women in the gumming and drying room", "Women operating machinery", and "Women perforating sheets of stamps." In these shots, machines take prominence in a space that is tight and visually crowded; industrial architecture is highlighted through the camera's angle and depth of view; women are prominently working machines, such as the steam-driven belts used for perforating stamps; and the repetition of spatial, human, and structural forms (including hood-like hats worn to protect against airborne dust) emphasizes a standardization associated with mass production and its forms of labor (Figures 22 and 23).

Spatial relations of machines and workers often distinguish Johnston's industrial images in the 1890s and 1900s. Two shots taken in a Lynn, Massachusetts, shoe factory that same year (1895) are careful to position women in relation to their machines and to include the surrounding array of machinery in the factory (Figures 24 and 25). The women in one photograph are literally framed by the machines, a relationship between woman and machine that Ridge's poem will evoke. "Woman Doing Bookbinding", taken as part of a series of women working at the Roycraft Print Shop, in East Aurora N.Y., frames a woman with her machine (Figure 26). In something of a contrast, other images from this 1900 series present a small business with a rather homey setting, suggesting a humane work environment that, while adopting prior conventions of representing women's work out of the home in domestic terms, offers an environmental view of labor that demands skill but does not dehumanize. "Seven Women Working in Roycroft Shop" captures the natural light filtering into the room, where women who are nicely dressed and comfortably spaced sit beneath house plants in an almost domestic architectural space; the domesticity of the small business, here visualized in positive terms as distinct from the factory, is also

apparent in an outside shot of the shop, where the workers are grouped cheerily on the front steps (Figures 27 and 28).

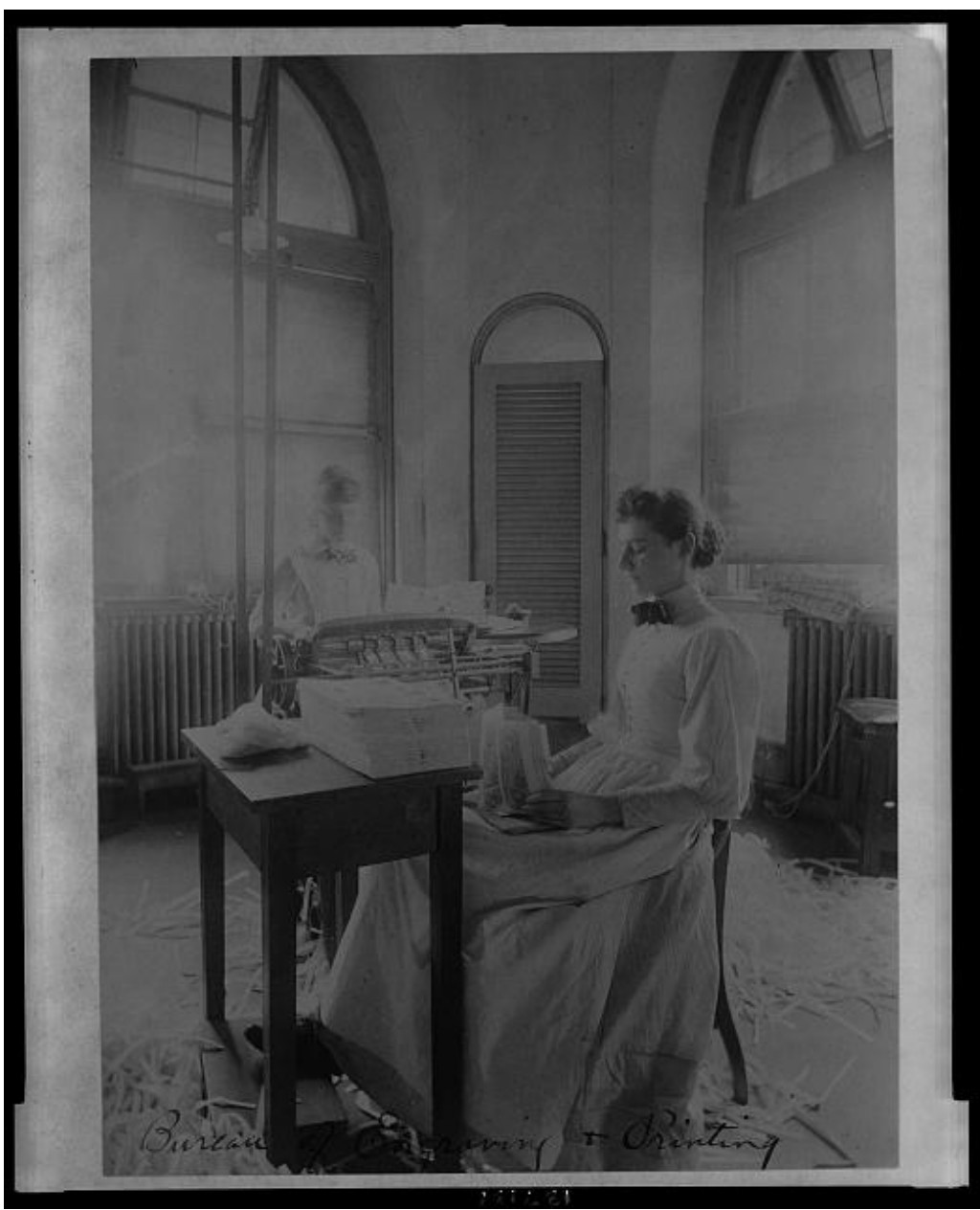

**Figure 20.** Frances Benjamin Johnston. 1890. Women trimming and stacking currency sheets at the Bureau of Engraving and Printing. Library of Congress.

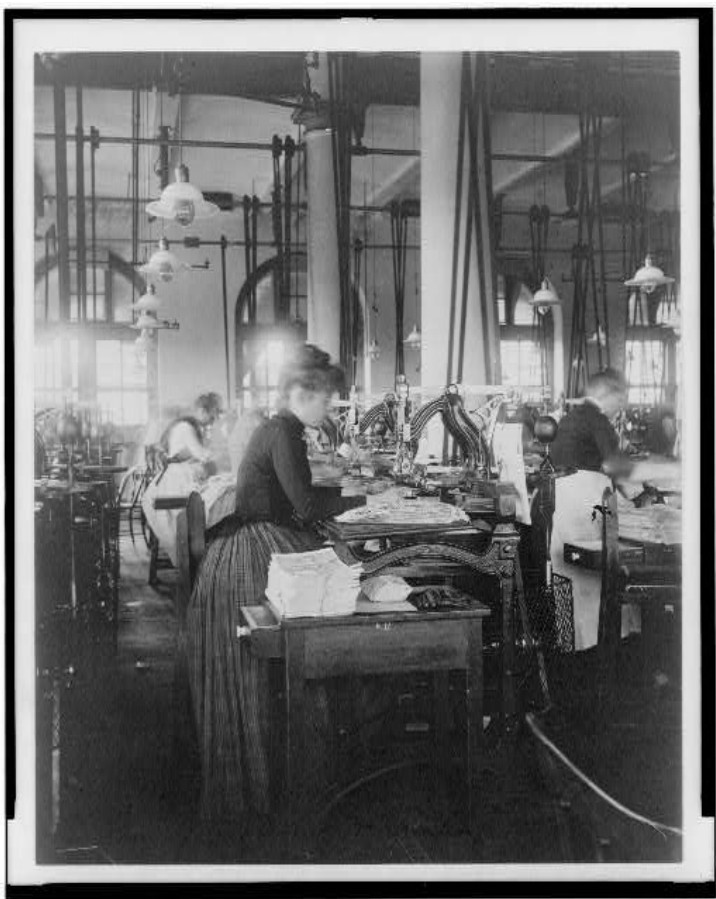

**Figure 21.** Frances Benjamin Johnston. 1890. Women operating machinery at the Bureau of Engraving and Printing. Library of Congress.

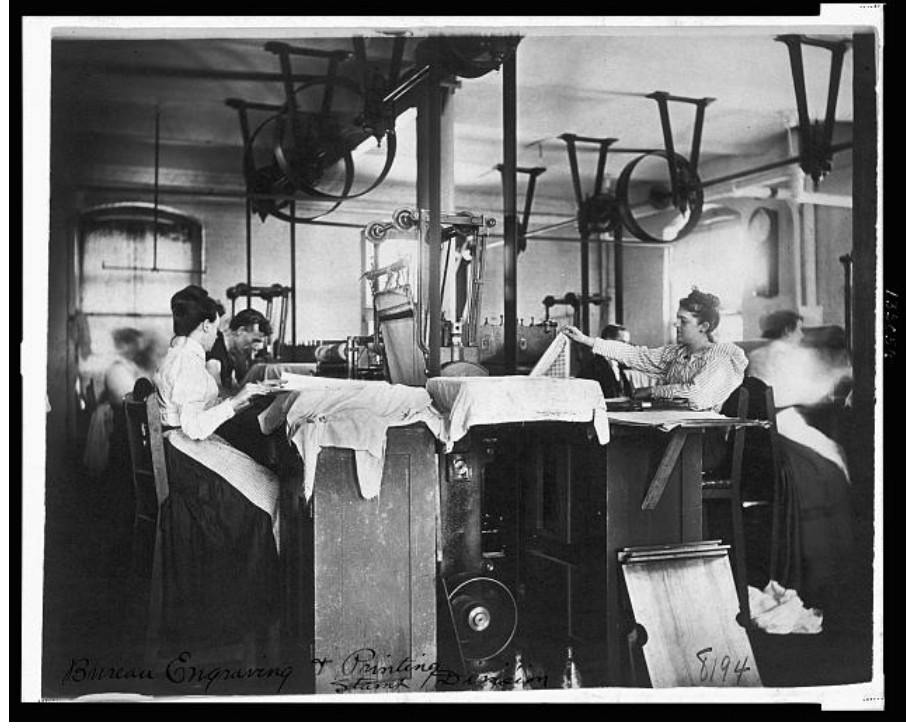

**Figure 22.** Frances Benjamin Johnston. 1895–1910. Bureau of Engraving & Printing—stamp division. Library of Congress.

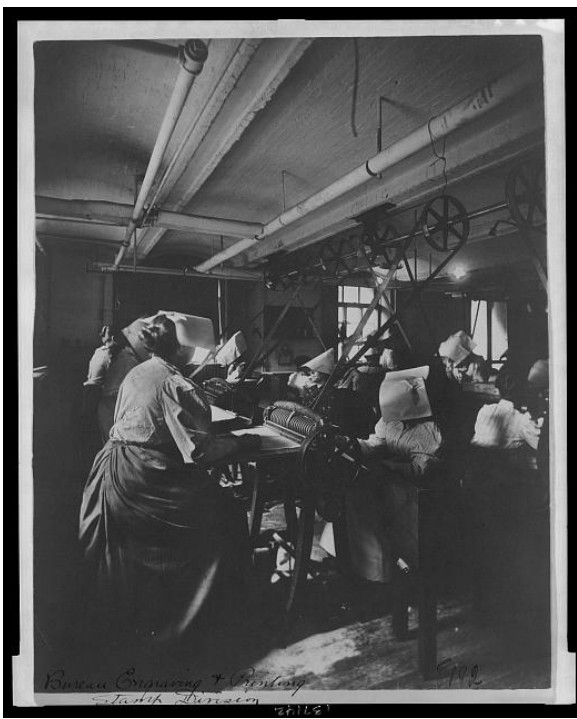

**Figure 23.** Frances Benjamin Johnston. 1895. Women perforating sheets of stamps in the Stamp Division at the Bureau of Engraving & Printing. Library of Congress.

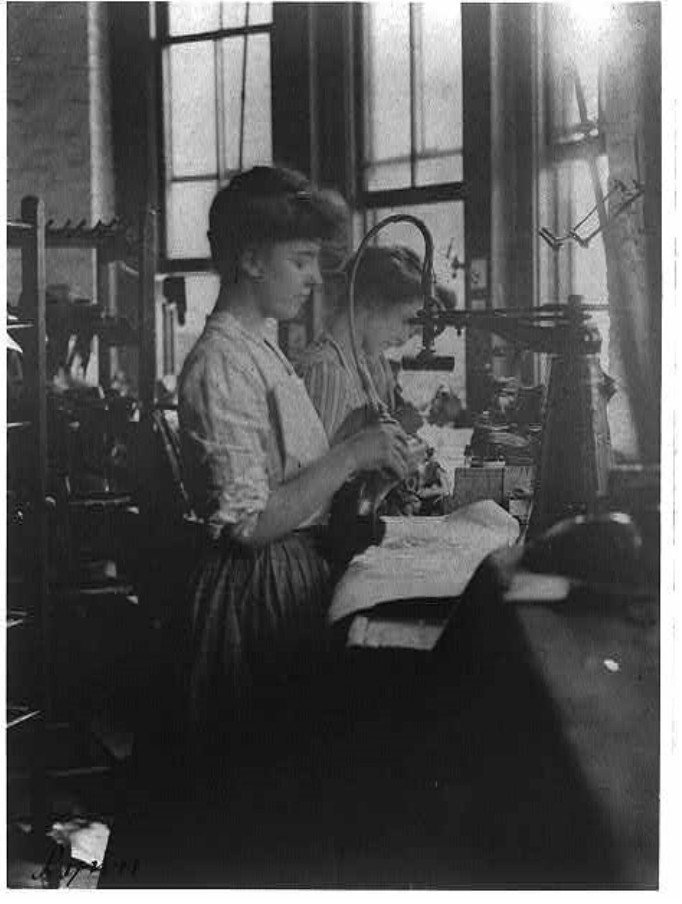

**Figure 24.** Frances Benjamin Johnston. 1895. Two women working in shoe factory, Lynn, M.A. Library of Congress.

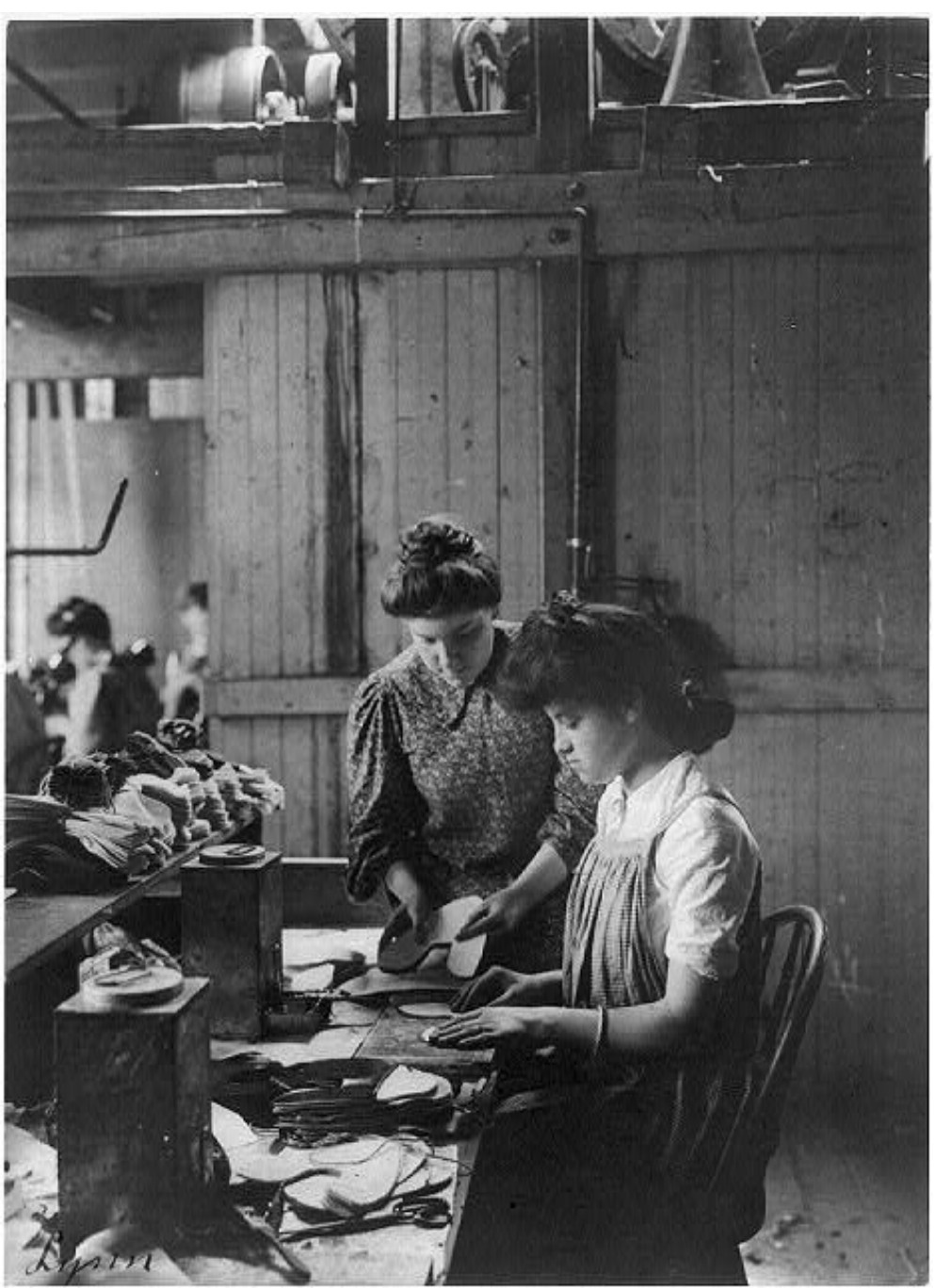

**Figure 25.** Frances Benjamin Johnston. 1895. Two women working in shoe factory, Lynn, M.A. Library of Congress.

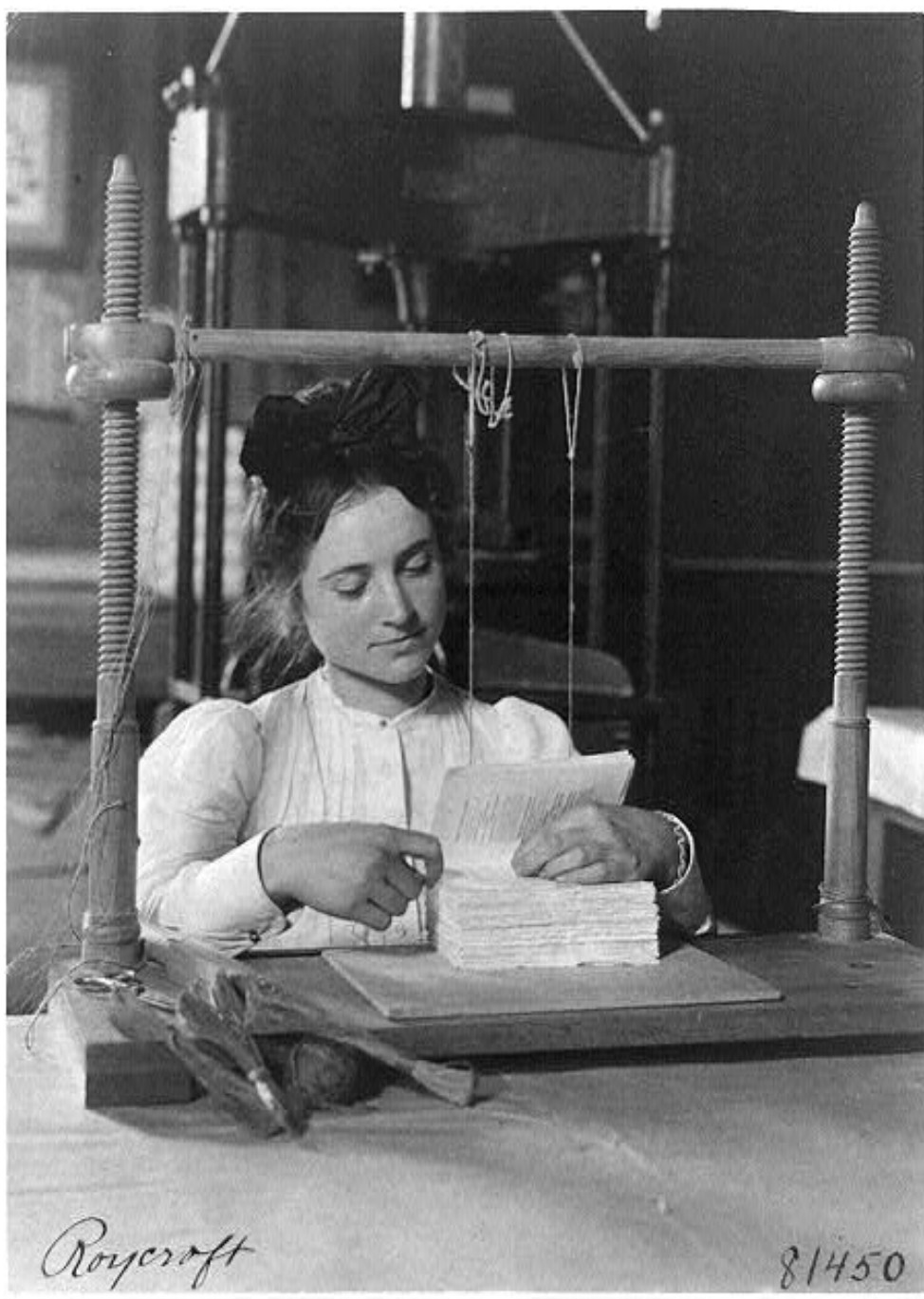

**Figure 26.** Frances Benjamin Johnston 1895. Woman Doing Bookbinding at Roycroft Shop. Library of Congress.

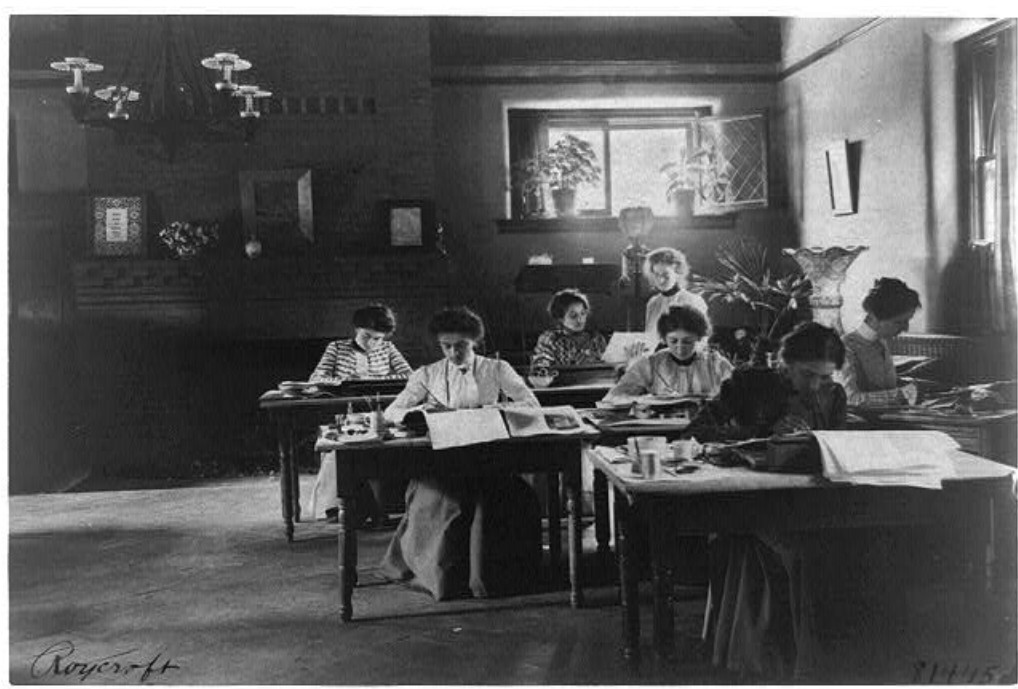

**Figure 27.** Frances Benjamin Johnston 1900. Seven women in the Roycraft Shop. Library of Congress.

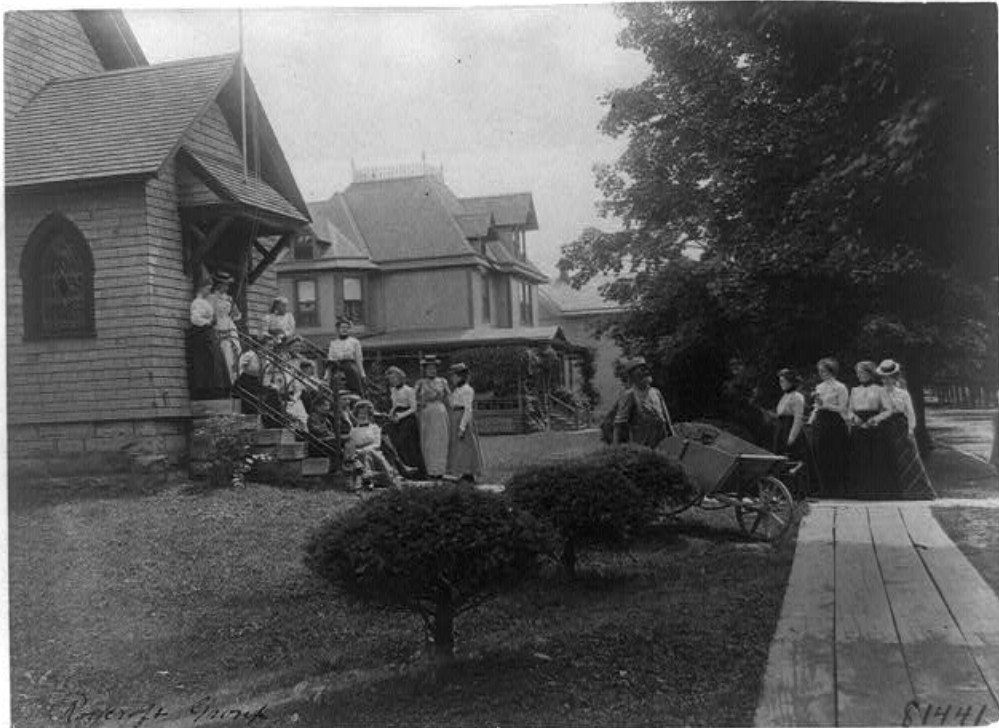

**Figure 28.** Frances Benjamin Johnson 1900. Women sitting outside the Roycraft Shop. Library of Congress.

Subsequently, and closer to her move to New York, a strikingly different 1910 shot of a wooden box industry dramatically adopts a severe perspective, composition, and viewing range that intensifies the inhumane associations of factory work (Figure 29). In a shot of the factory floor, an emphatically crowded space looms behind a worktable chaotically strewn with scraps, and Johnston aggressively foregrounds and foreshortens the table in the immediate picture plane. The composition then moves back through the factory space

into perspectival depth defined by sharp lines and angles. In this depth of space, repetition structures the scene: women are aligned visually with the architecture of production and products stacked like rows of columns, as if worker, machine, and product are all parallel in a labor system of mass production and standardization.

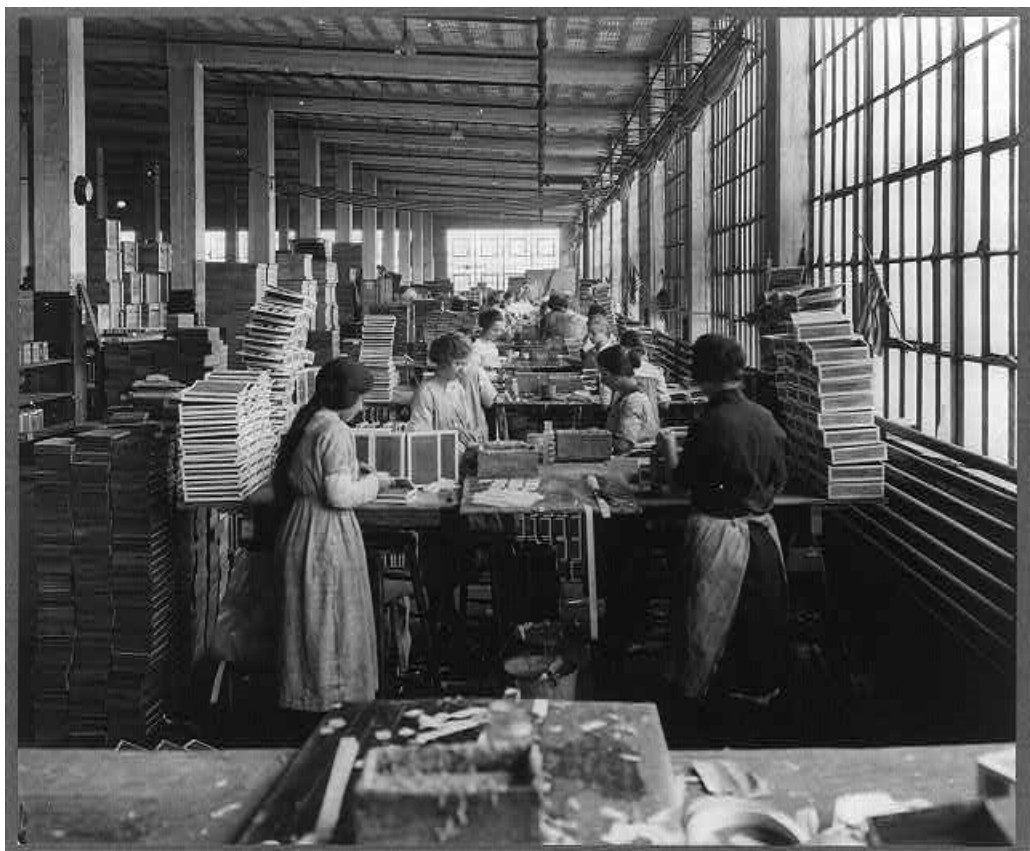

**Figure 29.** Frances Benjamin Johnston. 1910. Women in box factory.

Johnston's changing visual rhetoric moves from a pictorialist portrait in 1890 to an industrial document by 1910, suggesting a socially broader visibility of women workers, especially industrial workers, and a changing visual rhetoric for documenting working-class women within the context of their labor. In her 1910 box company photograph, Johnston uses composition and depth of field to suggest the devaluation of the worker and to emphasize the difficult and skilled labor of the women.

This image resonates with Ridge's images of Sadie, visualizing an environment in terms that could justify (to viewers) the necessity for resistance. Sadie's skillful battle with forces and systems of mechanized production refuses acquiescence to labor exploitation. Giving deliberate instructions to fellow women workers, she conveys her commitment to the class struggle and the agency of the worker. Drawing upon labor strategies popularized at the time, particularly the work slow-down, the poem suggests two related strategies as Sadie "bids the girls: 'Slow down –/you'll have him cutting us again!'" (Ridge 1918, p. 6). The reference to "cutting us" suggests a practice by which employers demanded speed for profit maximization but also would cut the rate paid for each piece as an employee reached or exceeded a predetermined amount. Refusing to use one's skill to reach that amount meant retaining a higher rate of pay for producing piecework goods. At the same time, in "an industrial system largely built on speed and profits", slowing the system thwarted profits and asserted the worker's power, however limited (Greenwald 1984, p. xviii). Sadie's words specify a specific form of labor sabotage that was increasingly encouraged, as when labor activist Elizabeth Gurley Flynn, speaking at the Paterson Strike on February 25, 1913 to some 24,000 workers, urged a slowing down or "withdrawl of efficiency":

> Sabotage means either to slacken up and interfere with the quantity, or to botch in your skill and interfere with the quality, of capitalist production or to give poor service. Sabotage is not physical violence, sabotage is an internal, industrial process. It is something that is fought out within the four walls of the shop. (Flynn 1913, np)

Reducing the quantity of goods produced, Flynn explains, "is a very old thing, called by the Scotch 'ca canny' . . . [or] the 'go easy' slogan, the 'slacken up, don't work so hard' species . . . . It is an attempt on the part of the worker to limit his production in proportion to his remuneration. That is one form of sabotage." Justifying sabotage as a form of power for the proletariat, "one weapon in the arsenal of labor to fight its side of the class struggle", Flynn insists that in "ca canny", laborers have "withdrawn their power as wealth producers from your plant and they are going to coerce you by this withdrawal of their power into granting their demands" (Flynn 1913, np).[22] Sadie's strategic call to fellow workers associates her with female leaders such as Flynn and connects the labor of women to a movement more often envisioned through a masculine, anti-capitalist virility.

Sadie's own power is visually and performatively pronounced, in contrast to prevalent notions, even among progressives, of working women as docile and acquiescent in the face of industrial exploitation.[23] Even as she calls to her fellow workers to slow down, she cannot resist marshalling her full degree of skill. As though impatient with restraints upon her skill and ability, or of being "Held in place by the fierce pressure all about—", she unleashes her power and again "Speeds up the driven wheels/And biting steel" (Ridge 1918, p. 6). The poem stresses both her high skill and strategic mind, as well as the frustrating necessity to limit one for the sake of the other.

Ridge's poetic perspective on industrial capitalism's impact—with its scientific work management theories regulating the body to enhance profit-making speed, heightened technology, and disregard for human well-being—shapes Sadie's image as a form of reportage documenting both dire conditions and women's capabilities. Such an approach bears affiliation with and suggestively reflects modern visual strategies encouraged by women's labor publications and early documentary photography, which helped shape the public's awareness of the conditions Ridge witnessed and provided a sense of visual fact or evidence.

## 4. Revising the Picture

Having shown Sadie in the factory, the poem turns to details of her social and private life, alongside her companions, Sarah and Anna. Paying particular attention to their clothing or manner of dress, their reading habits, and their love lives, these portraits register an awareness of cultural stereotypes casting working women as frivolous in consumption and shallow in self-edification while threatening in sexual promiscuity. Alternative to these largely middle-class anxieties about women working outside of the home, the poem presents these women's choices as both self-definitional and contributing to larger causes of justice and freedom.

Such anxieties attended the market's attention to working women's spending capacity. As Nan Enstad establishes in exploring cultural and social responses to the growing workforce of women, as "working women became an increasingly important market for both" fiction and fashion after 1870, such industries began to "target" this class of workers as consumers of low-quality goods (Enstad 1999, p. 21). Enstad convincingly argues that interests in fashion among wage-earning women added fuel to middle-class perceptions of questionable sexual virtue among such women, and that perceptions of "cheap fashion consumption made women themselves cheap, lowering their value and threatening their virtue even in the absence of sexual activity" (Enstad 1999, p. 30). Similarly, "middle-class women insisted that dime novels, like working women's fashion, lacked morality and taste" in their formulaic plots of romance and characterization of working-class heroines intended to appeal to the working-class woman (Enstad 1999, p. 41).

Ridge's attention to these areas of concern instead points to economic realities constraining consumption and, importantly, to strategic and self-aware choices in reading and relationships that situate the industrial worker as the modern new woman. As the poem moves from the factory scene to Sadie's home and social life, vocabulary echoing her adeptness with industrial processes stresses her commitment to intellectual labor. She reads at night "Those books that have most unset thought,/New-poured and malleable." The use of industrial terms that evoke amalgamating processes producing strong new substances such as steel suggests how Sadie's intellectual energy is a necessary and productive part of modernity's potential while counterpointing the mechanized shape of industrial labor.[24] Sadie's "thought/Leaps fusing at white heat" that fuels her revolutionary energy. Ridge's imagery suggests an intellectual understanding of politics, labor, and economy enabling women's activism and leadership capacity. Confident in her thinking, Sadie "spits her fire out" at a "protest meeting on the Square/Her lit eyes kindling the mob." The poem's integration of intellectual education and labor activism echoes efforts by women's trade organizations. *Life and Labor*, for example, included a regular section tellingly entitled "When We Have Time to Read." Short reviews of books, magazines, and pamphlets highlighted a range of national and international issues important to both the labor and women's movement (particularly linking the vote to labor equality) while emphasizing the importance of reading and exposure to new ideas for women workers.

Indeed, Sadie's reading habits are so voracious as to threaten her health, and she awakens "a little whiter . . . Alert, yet weary." Informed by her reading, her rebellion against the market's economic molding of both body and mind lessens her interest in so-called feminine concerns over how she looks. Mention of her clothing asserts her independence, as she "dresses in black" and seems unconcerned with fashion (Ridge 1918, p. 6). Sadie's tenement mate Sarah, who is "swarthy and ill-dressed," appearing "tousled and collar awry at her olive throat", works in a pants factory, but her real work is also in the mind, "hard and brilliant and cutting like an acetylene torch" (Ridge 1918, p. 8). As with Sadie, this industrial language insinuates Sarah is a worthy adversary of labor injustice and recasts the masculine image of the laborer typical in magazines such as *The Masses*, redirecting force and virility to the female worker whose mind becomes integral to her labor. As she "reads without bias . . . Psychology, plays, science, philosophies", the speaker imagines that "out of this young forcing soil what growth may come—what amazing blossomings", layering a particularly feminine generative force onto the "hard . . . torch" of her capacious mind (Ridge 1918, p. 8).

Ridge's insistence on Sadie's voracious night-time appetite for radical literature and an "ill-dressed" Sarah who reads diversely with intellectual vigor suggests a deliberate challenge to the middle-class stereotypes of factory workers as shallow and unsophisticated, or to labor union prejudices against women.[25] Another fellow worker, Anna, is also "different", and in the quick glance the poem offers of her, she is the beauty, with the "appeal of a folk-song", attracting men even with "her cheap clothes", an admission of fashion interest that is nonetheless financially restricted to mass-produced "cheap" ready-wear (Ridge 1918, p. 8). The middle-class criticism of cheap fashion consumption as indicative of poor virtue is recast in relation to Anna as a market matter, as wage pay restricts a woman's ability to acquire finer things while her "appeal" persists without more expensive fashion.

Fashion, and its ostensible power to distract men through promoting sexual desire, elicited sexual suspicion from critics of women's growing independence in general. A 1913 *New York Times* article covered, for example, a prominent anti-suffrage leader speaking out against clothing worn in a recent suffragist parade on Fifth Avenue, saying that "the costumes worn by women in the suffragist parade in New York last Saturday were patently intended to make an appeal to sex and not to the reason of men." She is quoted as saying that "the sex appeal was flagrant" and that clothing such as the "sheath gown, the split skirt, the low-necked gown in the broad light of day, and the general bizarre effect which could be calculated to make to a man a distinctly woman's appeal. It all goes to show that sex is at the bottom of the suffrage disturbance . . . " (George 1913, p. 10).

Such fears of unbridled sexuality responded to women's demands for equal rights and treatment, whether it be for the vote or working conditions and equal pay. As increasing numbers of women entered wage labor, an unsettled social reaction associated women's economic participation with frightening sexual independence. A distinct rhetoric of victimization linked wage-earning women with images of the prostitute, forced by low pay and labor exploitation to sell her body to survive. Tracing the production of images of the working-girl-turned-prostitute in mass visual culture, Schreiber notes that "associations between wage work, slavery, and prostitution had been present in the minds of Americans throughout the postbellum period" (Schreiber 2011, pp. 112–13).[26] Moreover, the Progressive Era's urban reforms included anti-prostitution efforts that circulated the "motif of victimization" beginning to reappear; for example, "in progressive women's magazines of the day that made use of the prevalent fears regarding forced prostitution to indict patriarchal society's collusion in the traffic of women", such as the Chicago-based *The Progressive Woman's* "White Slave Number" (Schreiber 2011, pp. 111–12). Generated in reaction to the specter of work outside the home but also as arguments against the insidious greed of capitalism, these associations played out in visual images depicting the sexualized body of the working woman.

More sympathetic to images of working women's sexuality, Ridge's portraits of Sadie, Sarah, and Anna join alternative visual contexts informed by this web of associations. The artist John Sloan, whose illustrations appeared in journals Ridge would have known, studied with other members of the Ash Can School at the Ferrer School with Robert Henri. Drawing for *The Progressive Woman* and the Socialist magazine *The New York Call* before joining *The Masses* in 1912 (the latter two of which published Ridge), Sloan contributed over five dozen illustrations over four years (Kitch 2001, p. 87). His drawings often commented on "class and public sexuality", responding to ways in which "the phenomenon of the woman in public was still profoundly troubling to many Americans . . . especially when she was working-class."[27] Many of his drawings, cartoons, and graphics appearing in progressive journals of the time strikingly confront the popular conflation of the single working woman and prostitution while drawing upon elements of fashion and environment.

Sloan's drawing "The Bachelor Girl", an image "typical of numerous images of urban working-class women", appeared in *The Masses* in 1915 (Figure 30).[28] The illustration's title employs a term used "to designate unmarried women", synonymous with "New Woman" (Schreiber 2011, p. 90). The drawing shows a working girl dressing (or undressing) before her closet, in skirt and slip, her coat and hat on the bed signifying her movement into urban public space or work. Her appearance of domestic self-sufficiency challenges "the conflation of sexually independent women with commercial sex workers" common to public perception, contesting "fears over the virtue of the single, urban working-class woman" (Schreiber 2011, pp. 95, 101). In "The Ghetto", Ridge similarly adopts the speaking voice of a "bachelor girl", a voice that surfaces first to introduce an "I" in the second section as both a participant in the workforce (as was Ridge) and an observer. The speaker locates herself in the small room she rents, independent but domestically enfolded by the family boarding her: "I room at Sodos'—in the little green room/With Sadie/And her old father and her mother" (Ridge 1918, p. 5). This position authorizes the poem's observational stance and its portraits of working women.

In her observations, the speaker's attention to women's comraderie distills the feminized space of "The Ghetto", from the opening section's imagery of a collective pageantry of sisterhood to other references to smaller groups of young, independent women strolling the streets, calling to friends, and attending political meetings as they claim public spaces for their bodies and voices. Capturing a modern mobility for working women, this comraderie finds visual expression in two of Sloan's cover images for *The Masses*. A 1913 cover, "The Return from Toil", depicts a group of six women, arms linked and bodies held in postures of laughter and glee as they converse while advancing down the street (Figure 31). They wear mid-length dresses with hats and purses that signal a fashion consciousness and a delight in their jaunty appearance. Sloan's drawing contrasts with negative reactions in

the popular press to fashions worn by strikers and a general disapproval of wage women indulging in "pretty dresses" they could afford (Enstad 1999, p. 100). As Enstad argues, such attempts at pleasure disagreed with middle-class perspectives of impoverishment, in which the so-called "deserving poor" would not spend money on "finery"; indeed, such waste characterizes the undeserving poor, whose own flaws are to blame for their situation in this popularly American way of thinking about poverty. The working "women's own ways of making meaning within that poverty" through modes of companionship, self-fashioning, and forms of inexpensive fun—as this image seems to record—were either eclipsed in main-stream media presentations of women's hardships in poverty, or were disavowed as indicators of working women's frivolity and, even, irrationality (Enstad 1999, p. 100).

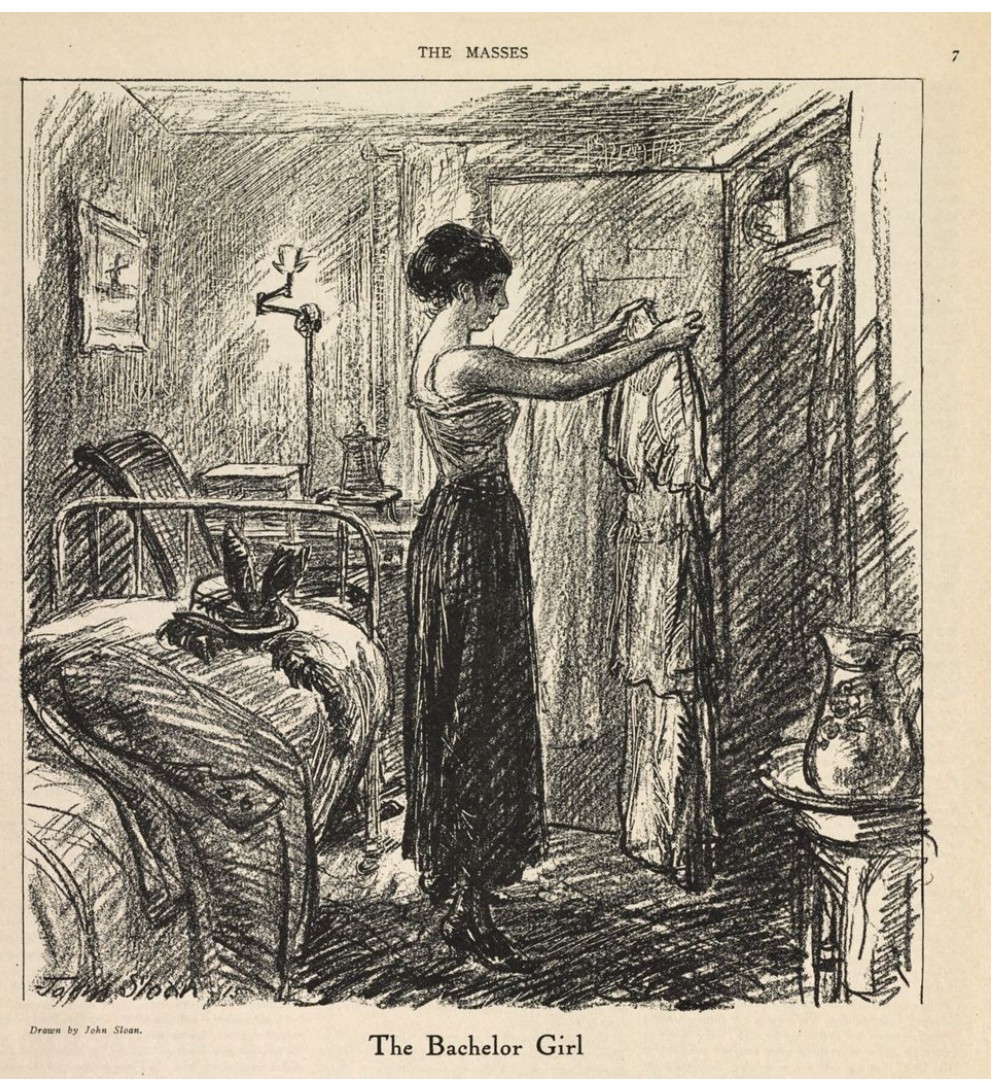

**Figure 30.** John Sloan. "The Bachelor Girl", in *The Masses* V6., No. 5. (February 1915), p. 7. The Modernist Journals Project. Available online: https://modjourn.org/issue/bdr527870/ (accessed on 29 August 2022).

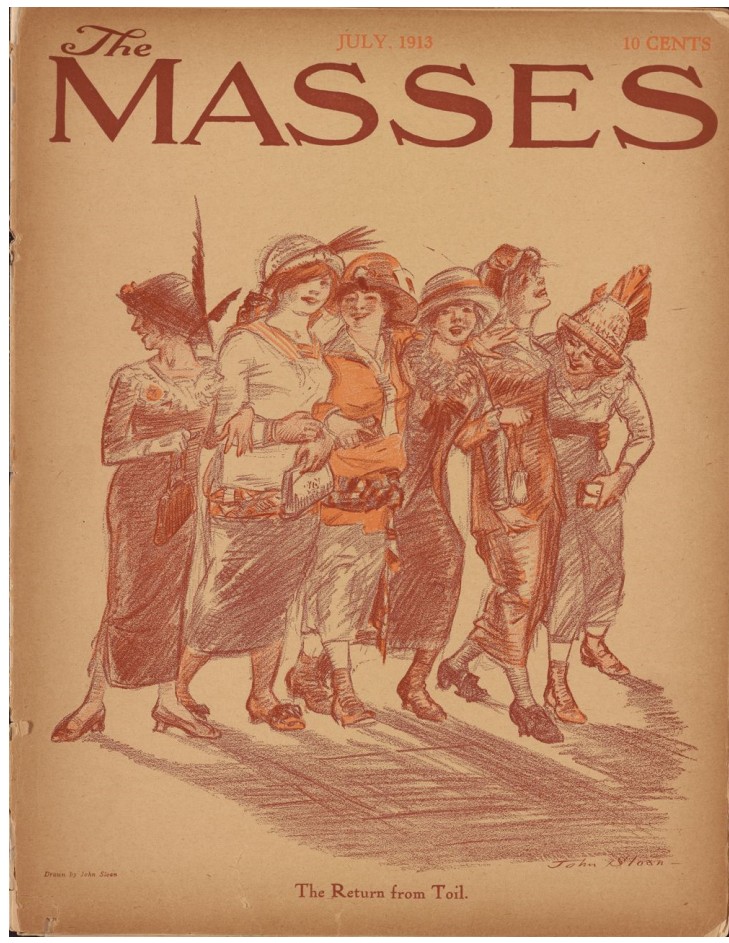

**Figure 31.** John Sloan. Cover image: The Return from Toil. *The Masses* V. 4 No. 10 (July 1913). The Modernist Journals Project. Available online: https://modjourn.org/issue/bdr527257/ (accessed on 29 August 2022).

These stereotypical attitudes mitigated the political agency of working women, especially of strike participants. Even sympathetic portrayals tended to present striking women as "proper charitable subjects" with "pathetic" lives, unlike the gleeful women Sloan presents in their after-work fashions. Indeed, Sloan's title pointedly references the conjunction of their "toil" with the visual depiction of what Ridge calls the "free comraderie" of "Young women [who] pass in groups" on the street, going to "forums and meeting halls" and calling "to the young men and to one another" on the street (Ridge 1918, p. 4). The imbrication of social gathering and political action presents a pleasure women have in each other's company while asserting their agency as laborers and activists intent on challenging the class structure. While Ridge's women bare their heads "to the stars", the hats worn in Sloan's image signify the prominence of trimmed hats (with lace, feathers, bows) as markers of both suffragist and labor activism (Figure 32). Hats could be trimmed inexpensively, thus even a cheap hat could be dressed up, allowing working women to both appropriate and undercut "middle-class efforts to control the definition of 'lady'" (Enstad 1999, p. 10). Suffragists similarly held hat-trimming contests for the best suffrage parade hat, in which no contestant could spend over a small sum on the hat, as ways to democratize the category of "woman" and to encourage both unity and self-creation among suffragists.[29]

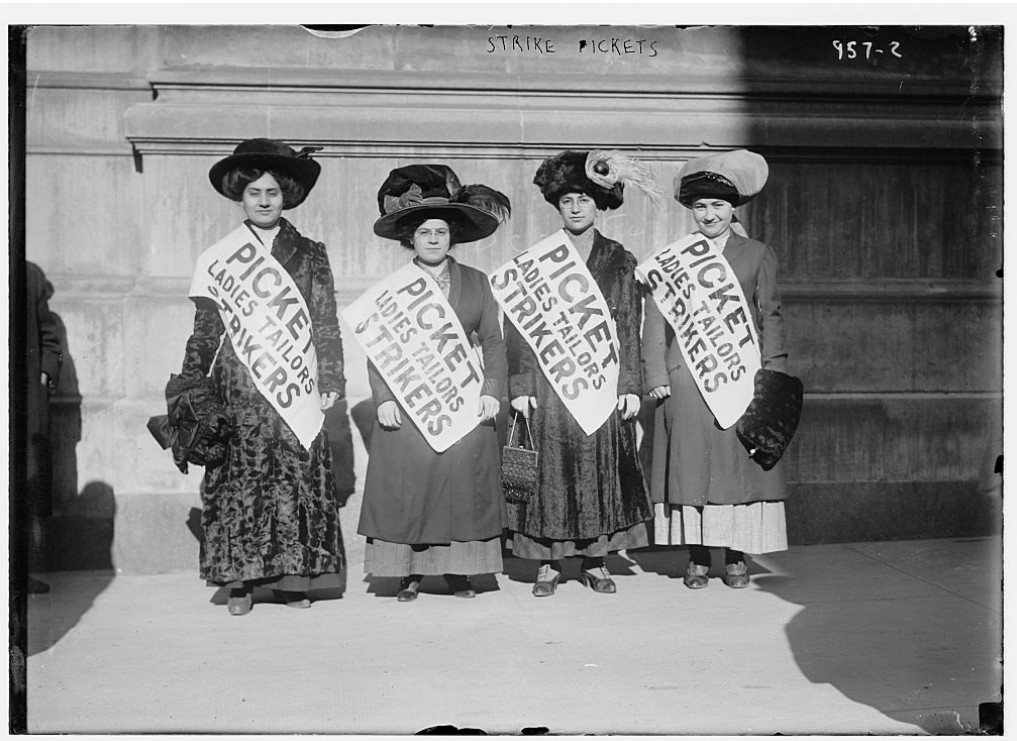

**Figure 32.** Bain News Service. 1910. Strike Pickets. Library of Congress.

In Sloan's depiction, and arguably in Ridge's explicit mention of Anna's inexpensive but pretty clothes, cheap fashion asserts independence and agency. In a 1913 cover, "Innocent Girlish Prattle—Plus Environment", Sloan ironizes public perceptions viewing working women's poverty and attraction to feminine fashion as incompatible (Figure 33). Presenting two young women, again with arms linked, strolling down a city street in attire that is both feminine (one woman's white dress has blousy ruffles and a sash that recalls suffrage costumes) and independent (the other woman wears a shirtwaist with a tie above a slender skirt, evoking the new woman), the image suggests the enjoyment they find in each other's company while depicting the hardships surrounding them. The night-time street scene of the title's "Environment" (suggesting the environmental view of poverty being argued at the time) includes a streetcar, men sitting on the curb, a heavy-set woman with arms crossed in front of a shop, and a sketchily rendered figure suggesting a beggar seated and bowed over beneath the shop window. The girls "prattle", a tongue-in-cheek reference to the perception of working girls as frivolous contrasted by the image's clear sense of their mobility and self-possession on the urban street. Like Ridge's working women, claiming their own sexuality and independence, this pair demonstrates what Schreiber describes as Sloan's "sympathy for the economic hardships faced by single urban women, who were staking new economic and sexual freedoms" (Schreiber 2011, p. 102).

Indeed, the poem's quick glances at Sadie, Sarah, and Anna's fashions join frank acknowledgment of their sexual energies. Sadie's mother, who hears the footsteps of her daughter's "Gentile lover" on the steps each night and the "soft babble of their talk", is more bothered that he is not Jewish than with Sadie's sexual freedom. Sarah's socialist-anarchism briefly but suggestively emerges in conjunction with the free-love movement, as her "desire covets nothing apart" and she "would share all things . . . /Even her lover" (Ridge 1918, p. 9, ellipses in text). Anna makes different choices. While she participates in labor activism and union strikes, giving up half her pay, she will not enter into the exchange that men might desire and expect, for she will "give anything—save the praise that is hers/And the love of her lyric body", suggestively alluding to women's autonomy and independence in relation to the larger male labor movement that had excluded them (Ridge 1918, p. 9).

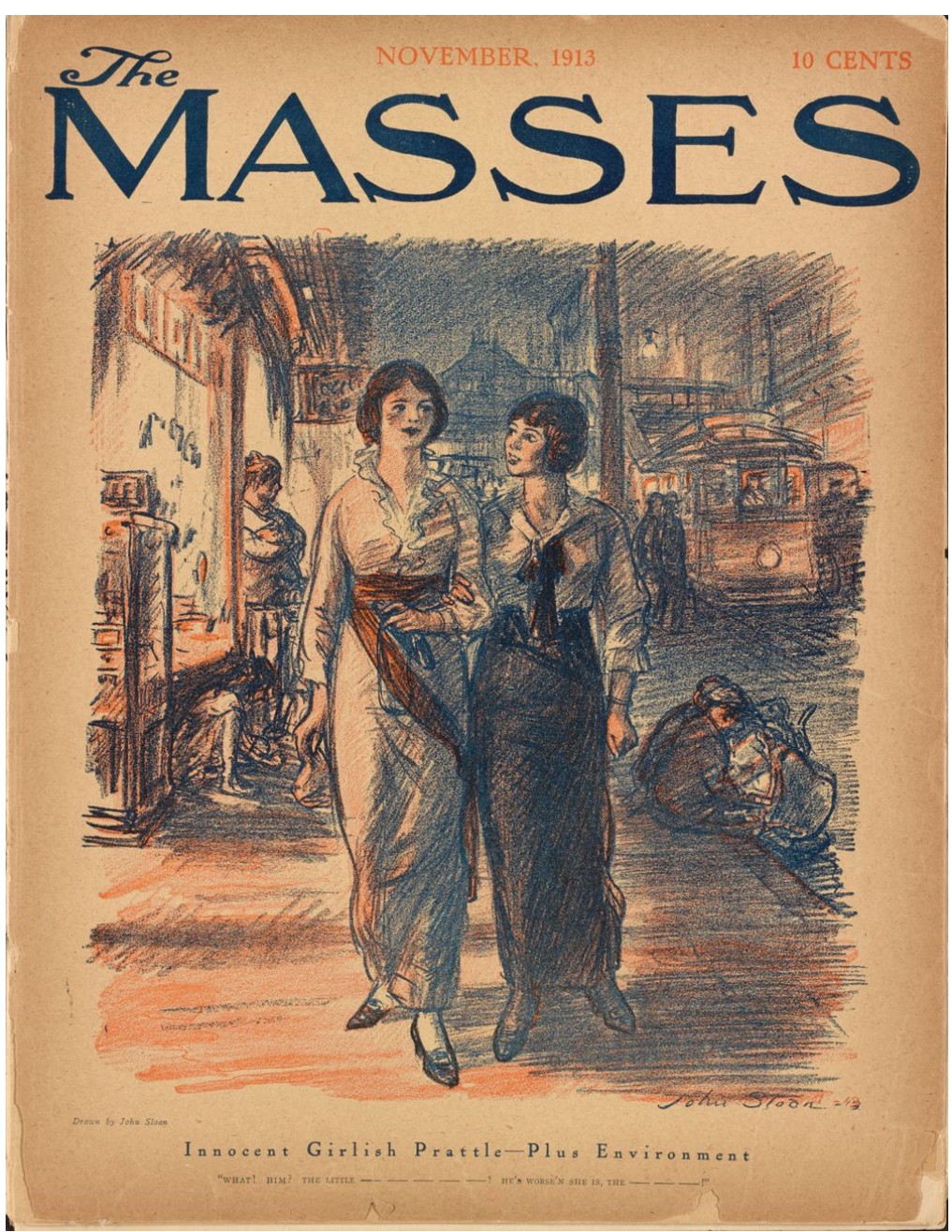

**Figure 33.** John Sloan. Cover image: Innocent Girlish Prattle—Plus Environment. *The Masses* V. 5. No. 2 (November 1913). The Modernist Journals Project. Available online: https://modjourn.org/issue/bdr527495/ (accessed on 29 August 2022).

In her portraits of these three women and their larger social lives, Ridge asserts that the independent working-class woman amplifies the "free comradarie" among men and women, choosing to experience her sexuality without being rendered victim or devoid of virtue; rather, her sexual agency is part and parcel of her intellectual and physical capability in the modern workforce (Ridge 1918, p. 4). Sadie's adoption of a "lover" of her choice, Sarah's willingness to "share everything", and Anna's control of her "lyric body" reverberate across several levels of labor discourse and visual rhetoric of the time. The poem's call to women's autonomy in the labor movement embodies women's agency in the fight for economic, social, political, and sexual justice. In "The Ghetto", Ridge's portraits celebrate the sexualized body as signifying modern choices for women to be celebrated, taking a position in opposition to much public opinion.

At a time of heightened labor activism by women in American cities, the poetic portraits of women in "The Ghetto" challenge the elision of modern women from popular visual iconography of modern labor, while also disputing stereotypes of women workers as victims or, in a more misogynistic vein, as irrational, silly, and inept in the modern workplace. As with the activists associated with industrial feminism, Ridge's portraits of women workers within a capitalist system are situated in relation to human need and a concept of justice predicated upon better conditions, wages, and quality of life—all made secondary if not invisible within the profit motive of industrial capitalism. Moreover, developing a distinctly political and gender-aware approach to modernism's imagist poetics, Ridge's first book suggests a culturally saturated notion of the image and its rapid dominance in modern culture in poems that register the power of visual imagery to persuade, indoctrinate, condemn, or expose. The visual image, for Ridge, is not neutral. While it cannot be separated from the cultural imperatives it serves (such as sexist ideologies), visual imagery can infuse a counter-point, serving to dismantle habitual ways of seeing and experiencing the world's material contexts. Ridge's poetic imagism engages its culture in this way, registering and deposing the power of visual imagery through engaging its discursive properties and changing the conversation, visually speaking, in a media-driven culture increasingly dominated by the visual image.

**Funding:** This research received no external funding.

**Conflicts of Interest:** The author declares no conflict of interest.

## Notes

1  See (Miller 2007, pp. 456–57) for a useful overview of depictions of the Lower East Side and its immigrant populations in the first two decades of the century.

2  See Kinnahan (2012) for a fuller discussion of economics in Ridge's poetry.

3  Tobin (2004, p. 65). On Ridge's editorial work with *Broom* and her literary salons, see (Tobin 2004, pp. 69, 76–78); for in depth discussion of Ridge's editorial involvement with *Others*, see Churchill (2006).

4  I am indebted to Berke's ground-breaking work on Ridge. Additional critics attuned to Ridge's visual poetics included NewcombeNewcomb, who discusses the visual language of the skyscraper; and Tobin, who considers the poem's structure as akin to the public "mural" (73).

5  As Miller (2007) notes, it was customary by the 1910s for literary writers, artists, intellectuals, and political and social activists to intermingle in New York's Lower East Side in circles that regularly included immigrants.

6  See (Orleck 1995) for an excellent history.

7  Van Wienan (2002, p. 192). The poem's epigraph specifies the occasion of its composition. "Bread" appears in full in Van Wienan anthology of American poetry of World War I. All quoted material from the poem refers to this edition.

8  Sloan, whose depictions of working class women will be discussed later, painted a "90-foot backdrop that dramatized the plight of many strikers" (Svoboda 2016, p. 79).

9  Ridge left New York in 1912, returning to live there again in 1917, although her activities with labor organizations across the country suggest that she remained up to date on New York's leading labor role while she continued to write about the city in those years.

10  Schreiber (2011, p. 51). See Schreiber's study for an excellent discussion of the gendering of work images at this time, particularly in the *Masses* but also across other publications.

11  See (Orleck 1995), for a detailed history of industrial feminism. See Vapnek (2009) for a history of labor collaborations of feminists, labor activists, and suffragists.

12  Vapnek (2009, p. 100). Vapnek details how such attitudes caused concern among middle-class consumers over the possible spread of contamination from goods made by these groups, especially those produced in home piece-work environments. The 1897 institution of the "white label" for factory-produced goods theoretically assured consumers of sanitary conditions of production, while also spelling the demise of home piece-work and encouraging the growth of factory manufacturing of clothing.

13  In (Orleck 1995, p. 107), emphasis added; originally in the *New York Call* 2 May 1914. The *Call* was a leftist publication familiar to Ridge and publishing her poetry, including "Bread."

14  *Life and Labor* (1911, V. 1.1, p. 3), unsigned opening page of first issue.

15  The *New York Times* reported different figures, stating on January 10, that 14,000 women across the garment industry were striking.

[16]　The development of the magnesium-cartridge pistol lamp that illuminated interior light shots made possible alarming and unposed depictions of living and working conditions, enhancing their aura of authentic documenation. Although his methods included barging in on his subject's homes unannounced and the book's descriptions of racial and ethnic groups are often problematically stereotypical, his images visualized a poor working class in ways that prompted substantial reforms in housing and labor conditions.

[17]　In 1908, Hine began work for the National Child Labor Committee, documenting child labor and advocating for reform through his images of children in factories, mines, mills, farms, and street trades, many of which circulated widely in newspapers, magazines, and posters, proving influential in changing public opinion and legislation regarding child labor.

[18]　Berke (2010, p. 31). Berke uses the terminology of the "snapshot" emerging at this time to stylistically describe the long poem.

[19]　I have found no evidence that Ridge and Johnstone knew one another, although their locations in Manhattan and their bohemian associations might well have led them to cross paths. See Berch (2000) for a full biography and Guimond (1991), who discusses the *Hampton Album*.

[20]　Johnston's photographs cited here are digitally available at the Library of Congress.

[21]　Published in "Uncle Sam as a Stamp-Maker", *Harper's Round Table* V 16 no. 815, June 11, 1895.

[22]　Flynn's full speech can be found at the Industrial Worker website: https://industrialworker.org/our-history-elizabeth-gurley-flynn-and-the-paterson-silk-strike/ (accessed on 29 August 2022).

[23]　Greenwald (1984, p. xxii). For example, in her introduction to Butler's ([1984] 1909) *Women and the Trades*, Greenwald notes how Butler assumes this passive image of women workers, despite her study's progressive aims in advancing better conditions and pay for them.

[24]　For a discussion of poets' varied responses in this era to industrial technologies, especially steel production and its importance in building the modern city and creating future possibilities, see Newcomb (2003) p. 104.

[25]　Enstad argues for an acknowledgement of working class women's practices of consumption of cheap fashion and dime-novel fiction as modes of imaginatively constructing identities as workers and as American.

[26]　Schrieber 111. Schreiber specifies that the term "white slavery" was "[i]nitially a phrase that designated anyone who was not in control of the means of earning one's living" but "by about 1905 the term had shifted to mean specifically women forced into prostitution" and "came to stand simply for the innocent, young, white female victims of the traffic in women"(113).

[27]　Kitch 88. See also Schreiber for a discussion of Sloan's use of a "journalistic style to pursue his fascination with working-class women and prostitutes" (92).

[28]　*The Masses*, 6. 5, February 1915, p. 7. https://modjourn.org/issue/bdr527870/# (accessed on 19 August 2022).

[29]　See, for example, Prize for Suffrage Hats (1913), in which the rules specified that "women should trim the hats themselves and use the regulation forty-two cent parade hat."

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
