# Peer review of "Portraits of Working Women: Lola Ridge’s “The Ghetto” and the Visual Record"

_humanities, doi:10.3390/h11050117_

Round 1
Reviewer 1 Report
This essay offers an interesting set of connections among verbal and visual depictions of women workers in the context of 1910s labor, feminism, and immigrant communities. At present, the essay's persuasiveness and value are undermined by its structure. The relationship of figure and ground is unclear at many points; are we thinking about Ridge's "The Ghetto" in the context of pictorial images by Hine or Johnston or Sloan or others, or are we thinking about the visual representations in light of Ridge's language? Paragraphs begin with statements of claims about one or another aspect of Ridge, but slip either to different elements of the poem (e.g. starting with portraits, slipping to spaces) or from Ridge to an artist. I'd recommend restructuring the essay along the lines of a clearly stated and outlined argument, with the opening sentences of paragraphs making clear moves in that argument and with the substance of paragraphs organized coherently around the move announced in the opening sentence. I'd also recommend slowing down in the analyses of Ridge's language; there is a tendency here to move too swiftly from quotation to conclusion, skipping the analysis that links the two (e.g. p. 12, where such diction as "brood," surge," and "tide" relates complexly to the ranks and files of suffrage or Shirtwaist marchers, or p. 24, where it's not clear how we get to the conclusion's shift from artisanal creativity to the industrial mechanized body -- though that seems to be treated in a subsequent paragraph, hence the confusion). When the verbal and visual are being related to each other, try to capture the relationship precisely; diction like "chime with" (p. 29) does not provide sufficient detail about the specific relationship.
It might be valuable to lay out the structure of the argument as a whole more carefully in the early paragraphs, and then, in moments of metacritical narration, to step back at moments of key shifts in the argument to tell the reader where on that early map they are in the unfolding of the argument. An early precis of Ridge's poem would also probably be useful, so that readers are oriented in the poem as a whole before individual paragraphs undertake analysis of specific imagery, diction, or characters.
A couple of minor spelling errors (Maren, not Marion, Stange; Tim Newcomb, not Newcombe), easily fixed.
Author Response
I am grateful for this perceptive reading, which really helped me untangle some issues I was aware of but had not resolved. My revisions are rather substantial, and I hope the argument and its structure are clearer. Thank you to the reader!
In revising, I have done the following:
- Rewritten the introduction (and abstract) to clarify the argument, lay out major points, and provide an intro to Ridge and a brief overview of the poem.
- Rethought and made more specific the thesis to more greatly emphasize and investigate the print media that DID pay attention to women laborers, amidst a more general paucity of imagery across the board. I have specified several genres of print culture: women's labor publications, social & industrial documentary photography, and radical press periodicals.
- I have eliminated / cut material that seemed less relevant to the shift towar a more specifically delineated focus.
- I have developed the research & discussion of women's labor publications, especially to extend to the WTUL's magazine and to consider its cross-overs with social documentary photography.
- I have restructured the essay extensively reframed its major topics and directions.
- I have worked to develop more precise idea-oriented topic sentences and cohesive paragraphs.
- In several instances, I have either reorganized or extended close readings of the poem to orient the reader.
- I have revised to bring the "figure" and "ground" into clearer relation (I found the reader's remarks particularly helpful here). The discussion of the poem, I hope, frames the movements into contextual materials. I have identified the essay's approach to "thickened contexts", as the visual and socio-cultural contexts at times demand sustained treatment. At the same time, the poem and its relevance to these contexts is kept more consistently in view and, I hope, moves the argument forward.
- I have corrected typos, paying particular attention to names (embarrassing sloppiness on my part).
Reviewer 2 Report
Humanities: Portraits of Working Women: Lola Ridge’s ‘The Ghetto ‘and the Visual Record
Notes for correction/revision
Structural Comments:
Article does an excellent job in reading sections of Lola Ridge’s decidedly understudied American long poem “The Ghetto,” in tandem with modernist and documentary visual culture. It also takes up two significant elements of Ridge’s aesthetic focus: women’s contribution to American industry and female immigrant desires. The article contains significant and new information, and there lies its primary challenge. While I recommend publication of this essay, I want to make several suggestions for strengthening its presentation.
Author needs to provide a broader overview of “the Ghetto” and its content. (It’s twenty-plus pages and a hodgepodge of communal life in flux that includes work, activism, leisure, and generational discordance). Highlight poem’s nine-sections and cue in readers to specific sections that are germane to the article’s overall argument and visual culture comparisons. The poem is not so well known in American literary study and will be new to some journal readers. Thus, a clearer singling out of specific sections would help guide them.
In connecting Ridge to Frances Johnston, a sharper distinction needs to be made between Ridge, a writer of modernist poetry, and Johnston, a “documentary” photographer. Ridge’s “snapshots” of women workers are part of an impressionistic, experimental literary interpretation of life in a Jewish ghetto. Although the poem is highly visual, and its author trained as a painter, it’s not a documentary work in the same manner that Riis, Hine, or Johnston’s photography is. While it’s clear that the author’s point is to discuss Ridge’s poetry along with, and in the framework of, “visual culture,” the poem requires a stronger statement that it is an experimental work of poetry and what that meant to a reading public in 1918. Succinctly establishing this fact at the outset would be useful.
The last section (4) has a lot of information in it. Ridge’s poem and its visual language, Johnston’s photographs, Sloan’s drawings, “The Masses.” It’s all good, interesting, but I think there needs to be clearer delineation between the visual “documentary” record – poetic characters as workers and activists, Johnston’s photos of women at work, information about industrial feminism--and the other “social” aspects such as fashion, consumption, and sex. These categories seem to float throughout the article and sometimes, out-of-bounds, which at times makes the comparisons difficult to follow. Sloan’s drawings and Enstad’s discussion of fashion and consumption need to be separated out more clearly from Johnston’s focus on the industrial aspect of women’s employment. Though nicely paired here, Johnston and Sloan suggest rather different aesthetic purposes from each other, and under the rubric of “visual culture” are also distinct in their purpose and design.
Section by section comments (corrections, suggestions, etc.)
Pt. 1 Introduction:
p. 1 line 32 – “pre-conversion” should be something like “before her conversion to Catholicism.” Younger readers may not know the trajectory of Dorothy Day’s radicalism.
p.2 line 67 -- Paul Avrich, who interviewed Lawson for his book “Anarchist Voices,” refers to Lawson as an “anarchist.”
p.2 line 70. East St. Louis, not East St. Lewis.
Pt. 2: Visual Print Culture
p. 3 footnote 3. Francisco Ferrer, not Edward Ferrer.
5, line 44. Perhaps add more from this comment from Kessler-Harris. Also consider looking atVapnik, Lara. Breadwinners: Working Women and Economic Independence. (Illinois, 2009). (Vapnik was AKH’s student).
10, line 297; the name is Maren Stange, not Marion Stange.
In general, the author rightly stresses that women were underrepresented in the radical press and labor journals of the era, but I think this point could be emphasized more. (a comment from the scholarship, some data, etc.)
Pt. 3: Ridge’s Visual Poetics
p. 15 line 435 – recommend stating “feminism’s first iteration,” rather than “first wave feminism.” The concept of “waves” is a contested one.
Pt. 4: Portraits of Working Women
Pps. 15-16: recommend looking at the Women’s Trade Union League’s journal Life and Labor, edited by Ridge’s Australian compatriot Alice Henry. (A digital version available or see Maciek’s 1986 overview of Life and Labor in History Workshop Journal). Life and Labor included guest columns and occasional pieces by immigrant women who wanted more representation in its pages and on the shop floor. Anyway, I think there needs to be more contrast between Johnston’s view of “industrial feminism” and Ridge’s poetic evocation of female workers and activists.
17, lines 512 . . . for comparison might note that, like Johnston, Ridge studied art. (Painting in Sydney before coming to America).
17. Footnote 12: Ridge wrote a number of long poems in various styles; “The Ghetto” is specifically a “modernist” or an “imagist” long poem. (One of these terms should be added).
27, lines 762-763: Not sure why Sarah is identified as a “southern European immigrant.”
There is no evidence that Sarah is ethnically different from the poem’s other Jewish female subjects, though the poem’s speaker does position herself as other.

Author Response
I am grateful for this perceptive reading, which really helped me untangle some issues I was aware of but had not resolved. My revisions are rather substantial, and I hope the argument and its structure are clearer. Thank you to the reader!
In revising, I have done the following:
- Rewritten the introduction (and abstract) to clarify the argument, lay out major points, and provide an intro to Ridge and a brief overview of the poem.
- Rethought and made more specific the thesis to more greatly emphasize and investigate the print media that DID pay attention to women laborers, amidst a more general paucity of imagery across the board. I have specified several genres of print culture: women's labor publications, social & industrial documentary photography, and radical press periodicals.
- I have eliminated / cut material that seemed less relevant to the shift towar a more specifically delineated focus.
- I have developed the research & discussion of women's labor publications, especially to extend to the WTUL's magazine and to consider its cross-overs with social documentary photography.
- I have restructured the essay extensively reframed its major topics and directions.
- I have worked to develop more precise idea-oriented topic sentences and cohesive paragraphs.
- In several instances, I have either reorganized or extended close readings of the poem to orient the reader.
- I have revised to bring the "figure" and "ground" into clearer relation (I found the reader's remarks particularly helpful here). The discussion of the poem, I hope, frames the movements into contextual materials. I have identified the essay's approach to "thickened contexts", as the visual and socio-cultural contexts at times demand sustained treatment. At the same time, the poem and its relevance to these contexts is kept more consistently in view and, I hope, moves the argument forward.
- I have corrected typos, paying particular attention to names (embarrassing sloppiness on my part).
- I have incorporated many of the smaller editorial suggestions regarding specific uses of language and clarification of distinctions.
Round 2
Reviewer 1 Report
I am grateful for the opportunity to read this revised version. What was already an interesting and well-researched piece of scholarship has, with the thoughtful and substantial revision the author has undertaken, become a sharp and cogent and utterly persuasive essay that at once grounds Ridge's poems in their historical contexts and shows how her poetic treatment of the figure of the laboring woman revises dominant cultural accounts (especially in contemporaneous visual culture). The argument is more clearly laid out in the early pages, the overview of "The Ghetto" enables a reader to be clearly oriented when passages are analyzed, and the analyses themselves more thoroughly develop the key points so that their connection to context and the contribution to advancing the argument are more legible. The essay is quite successful now in achieving its ambitious aims, and in bringing this particular kind of attention to Ridge's work it promises to renew our still developing understanding of this important poet.